# Glucose addition promotes C fixation and bacteria diversity in C-poor soils, improves root morphology, and enhances key N metabolism in apple roots

**Bianbin Qi[1,2], Kuo Zhang[1], Sijun Qin[1]\*, Deguo Lyu[1]\*, Jiali He[1]**

**1** College of Horticulture, Shenyang Agricultural University, Shenyang, Liaoning, China, **2** Academy of Agriculture and Forestry Sciences in the Great Xing'an Mountains, Jagedaqi, Heilongjiang, China

\* qsj1975@syau.edu.cn (SQ); lvdeguo@syau.edu.cn (DL)

**Data Availability Statement:** We have uploaded RNA-Seq data in the Sequence Read Archive (SRA) at NCBI under accession number PRJNA765206. We have also uploaded our research's minimal

## Abstract

The interaction between plant, soil and microorganism plays a crucial role in sustainable development of terrestrial ecosystem function and diversity. However, little information is known about how plant growth, soil organic carbon (C) fractions and microorganism respond to exogenous C addition in soils with low organic C content. Three levels of $^{13}$C-glucose (equal to 0, 100% and 500% of initial microbial biomass C) were added to non-sterilized (corresponding to treatment abbreviation of CK, Glu-1, Glu-2, respectively) and sterilized soils (corresponding to treatment abbreviation of SS, SS+Glu-1, SS+Glu-2, respectively) planted with apple rootstock (*Malus baccata* (L.) Borkh.) seedings. The objectives of this study were to analyse the dynamics of soil organic C (SOC) fractions and soil bacterial community diversity with glucose levels and soil sterilization, and to explore the morphology of roots and nitrogen (N) metabolism by plant after glucose addition to sterilized/non-sterilized soils. Results showed that the contents of labile organic C fractions were significantly varied (*P*<0.05) with the levels of glucose addition and soil sterilization. SS+Glu-2 and Glu-2 treatments increased the contents of labile organic C fractions, on average, by 48.47% and 35.33% compared with no glucose addition, respectively. About 21.42% and 16.17% of glucose-C remained in sterilized and non-sterilized soils, respectively at the end of experiment (day 45). Regardless of soil sterilized or not, the glucose addition increased the richness and diversity indices of soil bacterial community compared with no-glucose addition. The glucose addition optimized root zone conditions, and enhanced root vitality, morphology and biomass. Both SS+Glu-2 and Glu-2 treatments significantly enhanced (*P*<0.05) the contents of nitrate (NO$_3^-$-N) and nitrite (NO$_2^-$-N), but sharply decreased (*P*<0.05) the ammonium (NH$_4^+$-N) content compared with no glucose addition. Also, these two treatments significantly (*P*<0.05) increased the enzymic activities and gene transcript levels involved in root N metabolism, which demonstrated that the high level of glucose addition promoted N assimilation and transformation into free amino acids by root. Overall, the addition of exogenous C to not only promotes its fixation and bacterial community diversity in C-poor soils, but also improves root morphology and N absorption by plant.

underlying data set to the figshare with the doi: 10.
6084/m9.figshare.17913878.

**Funding:** Funded by (1) National Natural Science
Foundation of China (31972359) (2) Research and
Development Program of China
(2016YFD0201115) (3) the Agricultural Research
and Industrialization Projects of Liaoning Province
(2020JH2/10200028) (4) the China Agriculture
Research System of MOF and MARA (CARS-27).

**Competing interests:** The authors have declared
that no competing interests exist.

**Abbreviations:** C, carbon; SOC, soil organic
carbon; LOC, labile organic carbon; WSOC, water-
soluble organic carbon; POC, particulate organic
carbon; PE, priming effect; N, nitrogen; TN, total
nitrogen; $NO_3^-$, nitrate; $NO_2^-$, nitrite; $NH_4^+$,
ammonium; CK, Non-sterilized soil without glucose
addition; Glu1, Non-sterilized soil with low level of
glucose addition; Glu2, Non-sterilized soil with high
level of glucose addition; NR, nitrate reductase; GS,
glutamine synthetase; GOGAT, glutamate
synthetase; GDH, glutamate dehydrogenase; GOT,
glutamic oxalacetic transaminase; GPT, glutamic
pyruvate transaminase; OTU, operational
taxonomic units; C/N, Ratio of soil organic carbon
to total nitrogen; PcoA, principal coordinate
analysis; RDA, redundancy analysis; PGPR, plant
growth promoting rhizobacteria; SS, Sterilized soil
with glucose addition; SS+Glu1, Sterilized soil with
low level of glucose addition; SS+Glu2, Sterilized
soil with high level of glucose addition.

## Introduction

Soil is the largest carbon (C) pool in terrestrial ecosystems. The slight change in soil organic C
(SOC) can have a remarkable effect on global C budget and cycle. In addition, SOC is the main
component of soil organic matter, which has the advantages of supplying nutrients for crops,
increasing crop yields, and improving soil fertility and quality, etc [1,2]. Therefore, exploring
the mechanism of SOC dynamics is essential for sustainable development of agricultural and
environment. However, the change in total SOC is hard to be monitored in short term due to
its high background levels and the heterogeneous properties of soil [3]. Instead, labile organic
C (LOC) fractions in soil as sensitive indicators are used to describe the dynamics of SOC stor-
age. LOC fractions generally include water-soluble organic C (WSOC), microbial biomass C
(MBC), and particulate organic C (POC) [4]. Therefore, understanding the characteristics of
LOC fractions is an important to assessing SOC dynamics in agricultural ecosystems.

Exogenous C input is an effective measure to improve soil quality, promote nutrient
absorption, and increase crop production [5,6]. Exogenous C is transferred into LOC fractions
and stable SOC in short term [7]. However, LOC fractions show varied responses to exogenous
C addition. For instance, exogenous biochar has positive effects on MBC and POC in culti-
vated Brown soil, but shows negative effects on WSOC in the paddy soil [8,9]. Straw addition
increases or has no effects on soil LOC fractions [10,11]. Moreover, some research found that
the response of LOC fractions to exogenous C addition is distinctly varied among mineral fer-
tilizers supplies under rice-wheat system [12]. These divergent results should be associated
with soil nutrients, native MBC content, and quantity and quality of substrate addition. After
adding exogenous C to a soil with rich nutrients and high microbial activities, microorganisms
can be quickly activated from a dormant or starvation state. Thus, exogenous C addition
changes the contents MBC and SOC and influences the sequestration of exogenous C in soil
[13–15]. LOC fractions and SOC turnover are closely related with native MBC content. When
the amount of exogenous C addition is similar to MBC content, it induces a real priming effect
(PE), enhancing native SOC decomposition; when it is higher than MBC content, the PE
apparently promotes exogenous C decomposition [16]. Therefore, how LOC fractions of soil
respond to the levels of exogenous C addition is limited, especially under C-poor soil
conditions.

The supply of exogenous C substrate as microbial energy source increases soil microbial
biomass and further extends microbial activities [17]. Moreover, relative abundances of micro-
organism with different growth strategies are varied with soil nutrient environments, which
would shape different soil bacterial community structure [18]. In general, *K*-strategists have
lower growth rates and higher substrate affinities. Conversely, *r*-strategists have higher growth
rates, lower substrate affinities and preferentially assimilate labile C [19]. The supply of labile
C leads to the succession of microorganism from *r*- to *K*-strategists, and this process mainly
depends on nitrogen (N) captured by soil microorganisms [20]. On the other hand, the acti-
vated microorganisms acquire additional N sources from the mineralization of soil organic
matter under the absence of available N source, which could cause the competition between
soil microorganism and plant root for N absorption and utilization [21,22]. Their competition
is possibly regulated by bacterial community structure [23]. However, the specific taxonomic
groups, not all microorganisms, actively respond to exogenous C substrate [24]. As the compli-
cated soil backgrounds seriously interfere with the utilization of substrate by soil microorgan-
isms [25]. Given that soil sterilization strongly removes native soil microorganism and leaves
numerous empty niches for microorganism re-colonization, a new and activated bacterial
community structure would be formed [26,27]. The bacterial communities are primarily re-
colonized in sterilized soils and their structure turns to higher diversity and evenness during

recolonization session [28]. However, the dynamics of soil microbial communities after exogenous C addition to sterilized soil remains elusive.

N is an important and necessary element for plant growth and is a primary element for amino acids and proteins. Nitrate ($NO_3^-$) and ammonium ($NH_4^+$) are the main available N source of soil. After being taken up by plant roots, $NO_3^-$ and $NH_4^+$ are incorporated into glutamine and glutamate via N metabolizing enzymes, which are used to synthesize amino acids and nitrogenous compounds. N uptake by plant root is highly regulated by the addition of exogenous C to soil [29,30]. Previous research has demonstrated sugar effects on the N metabolism process in plants [31]. Moreover, lower level of sugar addition strongly inhibits $NO_3^-$ assimilation and decreases amino acid levels of plant [32]. Glucose is acted as an important signal molecule that regulates the genes expression of nitrate reductase (NR) [33]. However, little information is available on how the addition of exogenous C (such as glucose) affects N metabolism and genes expression concerned with N assimilation by plant root.

Apple is one of the principal fruit crops, due to high production capacity as well as economic value. China is of great importance in global apple production, ranking dominantly for planting area and fresh fruit exporting [34]. But more than half of apple harvested yields is less than 22.5 t ha$^{-1}$ [35]. Most apple orchards are usually established on hills or wastelands which have the characteristics of poor soil properties and low soil organic matter content (less than 1%) [6,36] Therefore, the interaction between SOC dynamics, root development and N metabolism, and soil microorganism is crucial to increasing SOC sequestration and apple yields.

The objectives of this study were to analyse the dynamics of SOC fractions and soil bacterial community composition and diversity with glucose levels, and to explore root morphology of apple and N metabolism by apple root after glucose addition to low C soils. We hypothesized that (i) Soil LOC fractions increased with the levels of glucose addition; (ii) Higher level of glucose addition increased bacterial community diversity and regulated the key genes involved in N metabolism activities. To address above hypotheses, we added three levels of glucose to non-sterilized/sterilized soils planted with apple rootstock, and determined the effects of glucose addition levels and soil sterilization on LOC fractions, soil bacterial community composition, and root morphology, amino acids, and key genes regulating N metabolism of root.

## Materials and methods

### Site description

In March 2019, soil samples were collected from a typical apple (Hanfu/*Malus baccata*) orchard in Xinmin, Liaoning, China (42˚ 4' 24"N, 122˚ 42' 41"E). The soil type is classified as Hapli-Udic Cambisol (FAO Classification). The altitude is 74 m, the average rainfall is 700 mm, and the average temperature is 7.6˚C in this orchard. The orchard was established in 2010 with conventional fertilization and field management. The distances of plant within and between rows were 2 m and 5 m, respectively. About 10 soil cores (40 cm × 40 cm × 40 cm) were sampled between rows of apple trees and sampling sites were about 2 m away from the tree trunks to avoid the interference of apple tree root system. The sampling sites were evenly distributed across the whole orchard to guarantee their representativeness. We picked out the visible plant root, rock pieces and the other debris from soil samples, passed them through a 2 mm sieve, and then fully mixed them for pot experiments. The basic properties of soil samples were as follows: a pH ($H_2O$) value of 6.5, 4.1 g kg$^{-1}$ soil organic carbon, 0.4 g kg$^{-1}$ total N, −18.3‰ δ$^{13}$C value, 188 mg kg$^{-1}$ MBC, 169 mg kg$^{-1}$ potentially available N, 11.1 mg kg$^{-1}$ available phosphorus (P), and 49.8 mg kg$^{-1}$ available potassium (K), and the percentages of sand, silt, and clay in soil were 72.4%, 26.7%, and 0.9%, respectively. The measured methods of SOC, total N, δ$^{13}$C value, and MBC were showed in the following section, and the contents of

available N, available P, available K, and pH value were analysed with the methods by Le and Marschner [37], and particle size separation was carried out with the method by Jensen et al. [38].

## Experimental design

A pot experiment was carried out in a greenhouse (12 h photoperiod, 500 μmol m$^{-2}$ s$^{-1}$ photosynthetically active radiation, 17–23˚C temperature, 55%-65% relative humidity) in Shenyang Agricultural University, Shenyang, China. Part of fresh soil samples was sterilized by autoclaving at 121˚C for 1 h, and then oven-dried at 40˚C for 2 days [39]. About 1 kg (oven-dried weight) sterilized/non-sterilized soil sub-samples were placed in plastic pots (internal diameter 10 cm, height 12 cm) and then the *Malus baccata* seedings (one plant per pot) with 6–7 leaves were transplanted into these pots.

The $^{13}$C-labelled glucose ($^{13}$C atom% = 99, Shanghai Research Institute of Chemical Industry Co. Ltd, Shanghai, China) was fully mixed with unlabelled glucose at a ratio of 1:10. The mixed glucose had a δ$^{13}$C value of 1789‰. High and low levels of mixed glucose at rates of 0.45 g kg$^{-1}$ and 2.25 g kg$^{-1}$ soil (equal to 100% and 500% of MBC, respectively) were dissolved in 100 mL distilled water and then the glucose solution was added to the non-sterilized soil (hereafter referred to as Glu-1 and Glu-2, respectively) and sterilized soil (hereafter referred to as SS+Glu-1 and SS+Glu-2, respectively) after the seedlings were transplanted for two weeks. The same amount of tap water and sterilized water (121˚C for 20 mins) were applied into non-sterilized (CK) and sterilized (SS) soils, respectively [40–42]. Thereafter, the plants in the sterilized and non-sterilized soils were supplied with sterilized water and tap water until harvest, respectively.

After glucose addition for 3, 7, 15, 30, and 45 days, soil samples were randomly collected from five pots with the similar seedling growth (one pot as one replication) per treatment. The aboveground seedings were firstly cut at the root base, and then the roots and soil cores remained in the pots were destructively collected. The soil samples adhered to root were carefully separated with shaking method because the seeding roots occupied the whole pots. After being removed the visible roots, the collected soil sub-samples were mixed thoroughly, and then were divided into half for further analysis. One half of sub-sample was stored at 4˚C for soil MBC and WSOC determination within 2 days. The remaining soil sub-sample was air-dried for total SOC and POC determination.

On day 45 after glucose addition, about 20 g homogenized fresh soil samples were collected and stored at -80˚C for microbial analysis. And 5 seedling roots were collected from each treatment, then were immediately frozen under liquid N$_2$. The frozen root samples were ground into fine powder with a ball mill (MM400, Retsch, Haan, Germany) and maintained at -80˚C for further analysis.

## Analysis for soil organic carbon fractions and total nitrogen

MBC was measured by fumigation extraction method [43]. The organic C contents of fumigated and unfumigated extracts were analysed using a Total Organic Carbon Analyzer (Element High TOC II, Germany) and adjusted using a conversion coefficient of 0.45.

WSOC was analysed using a modified method by Zhang et al. [44]. Briefly, fresh soil was added with distilled water at a ratio of 1:5, and shaken at 250 rpm for 30 minutes at 25˚C. The solution was centrifuged for 15 minutes at 3000 × g, and then the supernatant was filtered through a 0.45 μm membrane filter. The filtrate was measured by Total Organic Carbon Analyzer (Element High TOC II, Germany).

POC was determined depending on the procedure of Cambardella and Elliott [45]. Ten grams of air-dried soil was extracted with 30 mL $(NaPO_3)_6$ (5 g $L^{-1}$) and shaken at 150 rpm for 15 h. The soil suspension was filtrated through a 53 μm sieve. All materials remaining on the sieve were washed into a dry dish, then oven-dried at 75°C, and ground so as to measure organic C content. The total SOC, total N and POC contents were analysed with an elemental analyser (Elementar vario PYRO cube, Germany).

The contents of SOC and total nitrogen (TN), $\delta^{13}C$ values of SOC and LOC fractions were determined by an elemental analyser coupled with isotope ratio mass spectrometer (Isoprime 100 Isotope Ratio Mass Spectrometer, Germany). The $\delta^{13}C$ values were shown relative to Pee Dee Belemnite standard.

$\delta^{13}C$ value (‰) of MBC ($\delta^{13}C$ MBC) was calculated [46]:

$$\delta^{13}C_{MBC} = [(\delta^{13}C_{fum} \times MBC_{fum}) - (\delta^{13}C_{funum} \times MBC_{unfum})]/(MBC_{fum} - MBC_{unfum}) \qquad (1)$$

Where $MBC_{fum}$ and and $MBC_{unfum}$ are the amount of organic C (mg $kg^{-1}$) of fumigated and un-fumigated $K_2SO_4$ extracts, respectively; $\delta^{13}C_{fum}$ and $\delta^{13}C_{unfum}$ are the $\delta^{13}C$ values (‰) of fumigated and un-fumigated $K_2SO_4$ extracts, respectively.

Percentage of glucose-derived SOC in total SOC ($f_G$, %) was calculated [47]:

$$f_G = (\delta^{13}C_{SG} - \delta^{13}C_{S0})/(\delta^{13}C_{G0} - \delta^{13}C_{S0}) \qquad (2)$$

Where $\delta^{13}C_{SG}$ (‰) is the $\delta^{13}C$ value of SOC in the treatment with glucose addition; $\delta^{13}C_{S0}$ (‰) is the $\delta^{13}C$ value of SOC in the treatment without glucose addition; and $\delta^{13}C_{G0}$ (‰) is the $\delta^{13}C$ value of initial addition of glucose.

The residual percentage of glucose C in soil ($R_{glucose}$, %) was analysed [48]:

$$R_{glucose} = (C_{SG} \times f_G) \times 100/C_{G0} \qquad (3)$$

Where $C_{SG}$ represents the content of SOC derived from glucose C; and $C_{G0}$ represents initial glucose C content.

## DNA extraction, PCR amplification and bioinformatic analysis

Genomic DNA was extracted from soil samples at 45-day using a FastDNA Spin Kit (Omega Bio-Tek, Norcross, GA, USA) according to the manufacturer's protocol. The quality of DNA was analysed with 1% agarose gel electrophoresis and the total quantity of DNA was determined using a Thermo NanoDrop 2000 UV Microvolume Spectrophotometer (Thermo Fisher Scientific, USA). The primers 338F (5'-ACTCCTACGGGAGGCAGCAG-3') and 806R (5'-GGACTACHVGGGTWTCTAAT-3') were chosen to amplify the 16S rRNA genes in the V3-V4 regions [49]. The PCR amplification conditions included an initial denaturation at 95°C for 3 min, followed by 27 cycles of denaturation at 95°C for 30 s, annealing at 60°C for 30 s, extension at 72°C for 30 s, and a finial extension at 72°C for 10 min. The PCR products of all samples were purified with a Cycle Pure Kit (OMEGA), pooled in equimolar concentrations and performed on an Illumina (2 × 300 bp) MiSeq machine (Illumina, San Diego, CA, USA) at the Shanghai Origingene Biotechnology Co. Ltd., China.

The paired-end reads were analysed statistically by Trimmomatic software after depletion of primers. Bases of reads with a tail mass of 20 bp or less, overlapping paired-end reads less than 10 bp, and box sequences at both ends of reads were filtered. The unmatched sequences and singletons were excluded according to the Silva reference database v128 [50]. The operational taxonomic units were defined by clustering nonrepetitive sequences at 97% similarity and classified according to the Silva reference database using the Ribosomal Database Project

Bayesian algorithm classifier (RDP) [51]. Then, Usearch version 7.1 was used to cluster the sequences with 97% similarity for operational taxonomic units (OTU) [52].

Difference in the composition of bacterial OTUs according to taxonomic category between treatments was assessed. After centred-log ratio (*clr*) transformation (log transformation of the geometric mean), the '*codaSeq.clr*' function was used in the '*CoDaSeq*' package of R software [53]. The alpha diversity indices of bacterial communities, including ACE, Shannon and Simpson, were analysed using '*phyloseq*' package of R software. Principal coordinate analysis (PCoA) and redundancy analysis (RDA) were performed using '*stats*' and '*vegan*' packages in R software [54], respectively.

The raw sequences were deposited in the National Center for Biotechnology Information (NCBI) and Sequence Read Archive (SRA) number was PRJNA765206.

## Root surface, volume, total length and biomass

The surface, volume and total length of root were determined by using an image-analysis technique [55]. Roots were washed with distilled water and soaked in water contained in a transparent tray, then placed on Epson Perfection V800 Photo scanner (Epson, Long Beach, USA). The digitized images were measured by Winrhizo (Regent Instruments Inc., Quebec, Canada). The root was oven-dried for 24 h at 80˚C, and weighed with electronic analytical balance.

## Measurement of contents of nitrate, nitrite, ammonium and amino acids in root

Content of $NO_3^-$ in fresh root sample was determined with the method by Patterson *et al.* [56]. Contents of nitrite ($NO_2^-$) and $NH_4^+$ were used with the methods by Ogawa *et al.* [57] and by Bräutigam *et al.* [58] with some modifications, respectively.

Free amino acid extraction as described by Fürst *et al.* [59] with some modifications. About 0.3 g fresh root sample was extracted with 1 mL grinding media (deionized water/chloroform/methanol = 3/5/12, v/v/v). The extract solution was centrifuged at $12000 \times g$ for 15 min at 4˚C, filtered through a 0.22 μm organic membrane, and quantified by HPLC-MS/MS (Thermo Fisher Corporation, Waltham, Ma, UAS) with ESI source (Austion, Tx, USA). The HP $C_{18}$ column (4.6 mm × 150 mm, 5μm) was employed in a HPLC system. The flow rate was 1 mL min$^{-1}$, and column temperature was set at 50˚C. The mobile phase was made of ammonium acetate and acetonitrile (0.1% formic acid each). MS conditions were according to Jin *et al.* [60].

## Activities of key enzymes involved in N metabolism of root

About 0.2 g frozen root sample and 2 mL reaction agent (50 mmol L$^{-1}$ Tris-HCl with pH 8.0, 2 mmol L$^{-1}$ MgCl$_2$, 2 mmol L$^{-1}$ DTT and 0.4 mol L$^{-1}$ sucrose) were fully homogenized. The homogenates were centrifuged at $12000 \times g$ at 4˚C for 10 minutes, then the supernatants were used for the following analysis. The activities of NR and glutamine synthetase (GS) were measured according to Wang *et al.* [61] and Hageman *et al.* [62], respectively. Glutamate synthetase (GOGAT) and glutamate dehydrogenase (GDH) activities were assayed by monitoring the oxidation of NADH at 340 nm for 5 min and 3 min [63], respectively. Glutamic oxalacetic transaminase (GOT) and glutamate pyruvate transaminase (GPT) activities were measured by reacting enzyme extract with asparagine and alanine, respectively [64].

## Transcript levels of the genes involved in N metabolism of root

Total RNA from root samples were extracted using the Cetyltrimethyl Ammonium Bromide method [65]. The content and quality of extracted RNA were determination by

spectrophotometer (Nano Drop 2000; Thermo Fisher Scientific Ltd., New York, USA). The first-strand cDNA was synthesized by a 20 μL total volume using a PrimeScript RT reagent kit (DRR037A; Takara, Dalian, China) with the instruction of manufacturer's protocol. Quantitative real-time expression (qRT-PCR) on the genes was performed containing 10 μL of 2×SYBR Green Premix Ex Taq II (DRR820A; Takara, Dalian, China), 0.5 μL cDNA, and 0.2 μL primer. The detail designs for each gene were listed in S1 Table. The reaction was tested in a CFX96 real time system (CFX96; Bio-Rad, Hercules, CA, USA). β-Actin was the reference gene. PCR was conducted in five replications for each gene. Relative mRNA expression was calculated according to the $2^{-\Delta\Delta Ct}$ method [66].

### Statistical analysis

All data in figures and tables were presented as means ± standard error (SE). SPSS version 19.0 (IBM Software, Chicago, IL, USA) was used for all statistical analyses. Analysis of variance (ANOVA) followed by Duncan tests was conducted to analyse significant difference among treatments at $P < 0.05$.

## Results

### Contents of soil organic carbon fractions and total nitrogen

The addition of glucose significantly increased the content of total SOC by 1.7%~11.7% (S1A Fig). Compared with sterilized soil, non-sterilized soil with high level of glucose addition decreased the content of total SOC, on average, by 3.7%, while that with low level of glucose addition and without glucose addition increased the content of total SOC by 5.4%~7.5% during the whole sampling time.

The Glu-1 and Glu-2 treatments increased the MBC content, on average, by 14.9% and by 46.6% compared with CK treatment, respectively (S1B Fig). The content of MBC was 34.6% and 121.3% higher in the SS+Glu-1 and SS+Glu-2 treatments than that in the SS treatment during the whole sampling time. The WSOC content in the SS treatment was, on average, 6.5% higher than that in the CK treatment during the whole sampling time (S1C Fig). The SS +Glu-1 and SS+Glu-2 treatments increased the content of WSOC, on average, by 13.4% and 42.7% relative to SS treatment during the whole sampling time. The POC contents appeared gradually increased from 0 to 7 days, and then tended to be stable with sampling time (S1D Fig). The POC content was increased with the glucose addition levels. There was little variation ($P$>0.05) in POC content between SS+Glu-1 and Glu-1 treatments during the whole sampling time except for 15 and 45 days.

The glucose addition significantly enhanced the TN content of soil by 22.1% (S2A Fig). TN content was significantly increased ($P$<0.05) by 20.1% and 23.4% in Glu-2 and SS +Glu-2 treatments compared with Glu-1 and SS+Glu-1 treatments, respectively. The C/N ratio in Glu-2 and SS+Glu-2 treatments was 21.3% and 16.1% lower than that in CK and SS treatments, respectively. The C/N ratio was decreased with the levels of glucose addition (S2B Fig).

### δ¹³C values of total soil organic carbon and its fractions

The δ¹³C value of total SOC in SS+Glu-2 and SS+Glu-1 treatments was, on average, 10.1% and 12.8% higher than that in Glu-2 and Glu-1 treatments during the whole sampling time, respectively (S3A Fig). The δ¹³C value of SOC in treatments without glucose-C addition (CK and SS) remained essentially unchanged with sampling time.

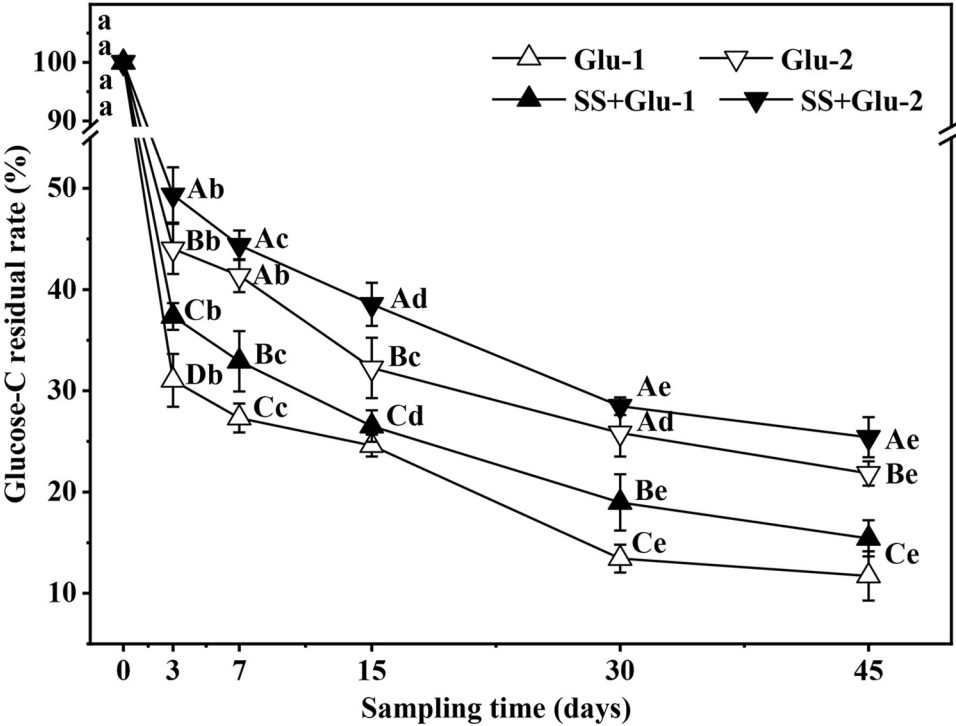

**Fig 1. Change in glucose C residual rate with sampling time in the sterilized and non-sterilized soils with glucose addition.** Different uppercase letters indicate significant differences ($P < 0.05$) among different treatments at the same sampling time. Different lowercase letters indicate significant differences ($P < 0.05$) among different sampling time within the same treatment. Overlapping date points with the same significant differences are indicated by common letters. CK, non-sterilized soil without glucose addition; Glu-1, non-sterilized soil with low level of glucose addition; Glu-2, non-sterilized soil with high level of glucose addition; SS, sterilized soil without glucose addition; SS+Glu-1, sterilized soil with low level of glucose; SS+Glu-2, sterilized soil sterilization with high level of glucose addition.

The $\delta^{13}C$ value of MBC in SS+Glu-2 and Glu-2 treatments was, on average, 39.5% and 31.5% larger than that in SS+Glu-1 and Glu-1 treatments during the whole sampling time, respectively (S3B Fig). The SS+Glu-1 and SS+Glu-2 treatments increased $\delta^{13}C$ value of WSOC, on average, by 31.1% and by 27.1% compared with Glu-1 and Glu-2 treatments during the whole sampling time, respectively ($P<0.05$, S3C Fig). The glucose-C addition increased $\delta^{13}C$ value of POC by 6.3%~15.8% during the whole sampling time (S3D Fig).

## Glucose C residual rate and net SOC balance

About 49.1%~33.3% of glucose C was remained in soil at 3 days, then presented a slow decline trend afterwards (Fig 1). At the end of sampling time (day 45), glucose C residual rate was 19.6%~24.2% in the treatments with low level of glucose addition, and was 28.1%~35.2% in the treatments with high level of glucose addition. The glucose C residual rate in SS+Glu-2 and Glu-2 treatments was, on average, 12.4% and 10.1% larger than that in SS+Glu-1 and Glu-1 treatments during the sampling time, respectively.

The net SOC balance equalled to the difference between native SOC decomposition and new SOC formation derived from glucose C (Fig 2). The formation of new SOC derived from glucose C almost equalled to the native SOC loss in the Glu-1 treatment. The fixed glucose-C in soil (0.08 g kg⁻¹) was enough to offset the native SOC loss (0.02 g kg⁻¹) in the SS+Glu-1 treatment. The content of net SOC balance was 0.11 g kg⁻¹ and 0.20 g kg⁻¹ in Glu-2 and SS+Glu-2 treatments, respectively.

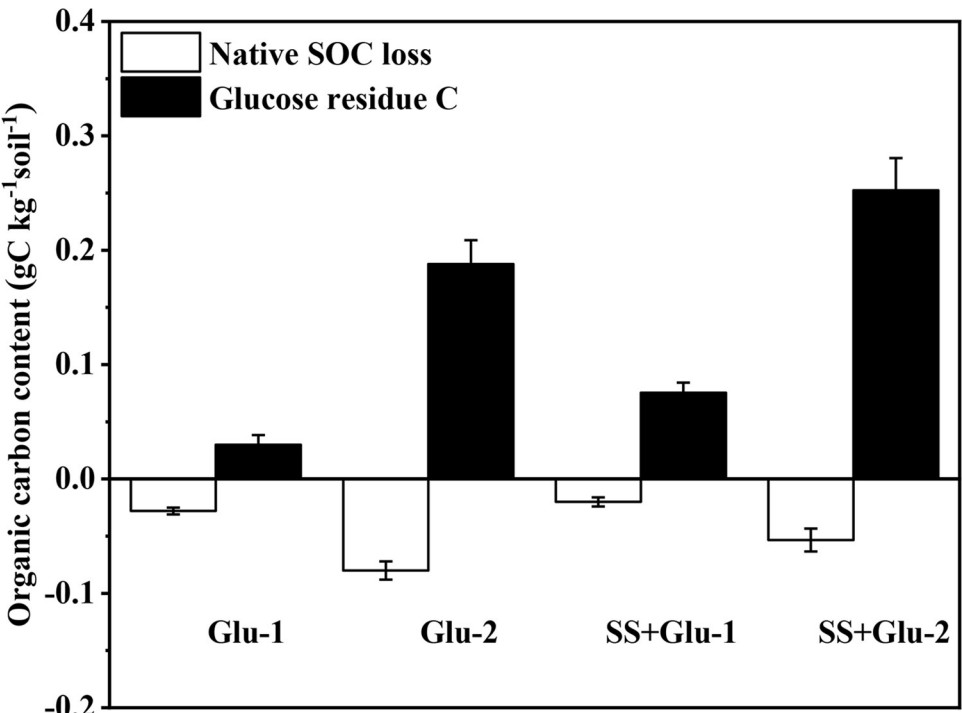

**Fig 2. Change in net soil organic carbon balance in the sterilized and non-sterilized soils with glucose addition at day 45.** CK, non-sterilized soil without glucose addition; Glu-1, non-sterilized soil with low level of glucose addition; Glu-2, non-sterilized soil with high level of glucose addition; SS, sterilized soil without glucose addition; SS+Glu-1, sterilized soil with low level of glucose; SS+Glu-2, sterilized soil sterilization with high level of glucose addition.

### Soil bacterial richness and diversity indices

The coverage, which was used to assess the sequencing quality, exceeded 0.994 in all treatments. The Ace, Chao, Shannon, and Simpson indices were used to evaluate the richness and diversity of soil bacterial communities (Table 1). The values of Ace, Chao and Shannon indices were increased with the glucose addition levels, whereas the Simpson index showed the opposite trend. Compared with non-sterilized soil, sterilized soil with high level of glucose addition increased the values of Ace, Chao and Shannon indices by 13.8%, 7.9% and 11.4%, respectively; and sterilized soil with low level of glucose addition increased them by 17.3%, 13.8% and 2.6%, respectively.

**Table 1. The alpha diversity indices of bacterial communities in the sterilized and non-sterilized soils with glucose addition at day 45.**

| Treatment | Ace | Chao | Shannon | Simpson | Coverage |
|---|---|---|---|---|---|
| CK | 1511.3±170.5 e | 1593.7±107.5 e | 5.58±0.06 e | 0.040±0.004 a | 0.995 |
| Glu-1 | 1811.1±137.1 d | 1806.3±118.2 d | 5.71±0.05 d | 0.034±0.002 b | 0.998 |
| Glu-2 | 3198.2±169.6 b | 3064.7±103.4 b | 6.23±0.03 b | 0.017±0.001 d | 0.994 |
| SS | 1755.7±155.3 de | 1627.6±120.1 de | 5.78±0.03 d | 0.032±0.005 b | 0.994 |
| SS+Glu-1 | 2124.1±142.7 c | 2055.5±133.9 c | 5.85±0.02 c | 0.026±0.001 c | 0.995 |
| SS+Glu-2 | 3639.7±137.4 a | 3306.0±125.2 a | 6.94±0.05 a | 0.020±0.001 d | 0.994 |

Different lowercase letters in the same column indicate significant differences between treatments ($P < 0.05$). CK, non-sterilized soil without glucose addition; Glu-1, non-sterilized soil with low level of glucose addition; Glu-2, non-sterilized soil with high level of glucose addition; SS, sterilized soil without glucose addition; SS+Glu-1, sterilized soil with low level of glucose; SS+Glu-2, sterilized soil sterilization with high level of glucose addition.

## Soil bacterial community composition

The PCoA plot based on the *clr*-transformed data was used to presented the changes in soil bacterial community structures. The first two principal components explained 78.9% of total variations in the composition of bacterial communities (Fig 3A). The PCoA1 clearly separated the treatments with and without glucose addition. The non-sterilized and sterilized treatments were differentiated along the PCoA2. As presented by the hierarchical cluster analysis, bacterial communities revealed two clusters comprising samples from all treatment groups (Fig 3B). The treatments of non-sterilized soil and sterilized soil with the same level of glucose addition clustered together.

The predominant bacterial phyla in all treatments were *Proteobacteria*, *Actinobacteria*, and *Acidobacteria*, with relative abundances larger than 10% (Fig 3C). The glucose addition enhanced the relative abundances of *Proteobacteria*, *Actinobacteria* and *Firmicutes* by 59.4%, 19.6% and 43.3%, but decreased those of *Acidobacteria* and *Chloroflexi* by 41.8% and 39.1%, respectively. The relative abundances of *Proteobacteria*, *Actinobacteria* and *Verrucomicrobia* were higher in the SS treatment than those in the CK treatment. Within the *Proteobacteria* phylum, the families *Bradyrhizobiaceae*, *Hyphomicrobiaceae* and *Rhodobiaceae* in the order *Rhizobiales* were in the SS+Glu-2 treatment higher than those in the other treatments (S4 Fig).

The heatmap of soil bacterial genera showed that all samples were clustered into two groups consisting of the treatment with glucose addition and that without glucose addition (Fig 3D). The relative abundances of members of *Proteobacteria* (*Pseudomonas*, *Skermanella* and

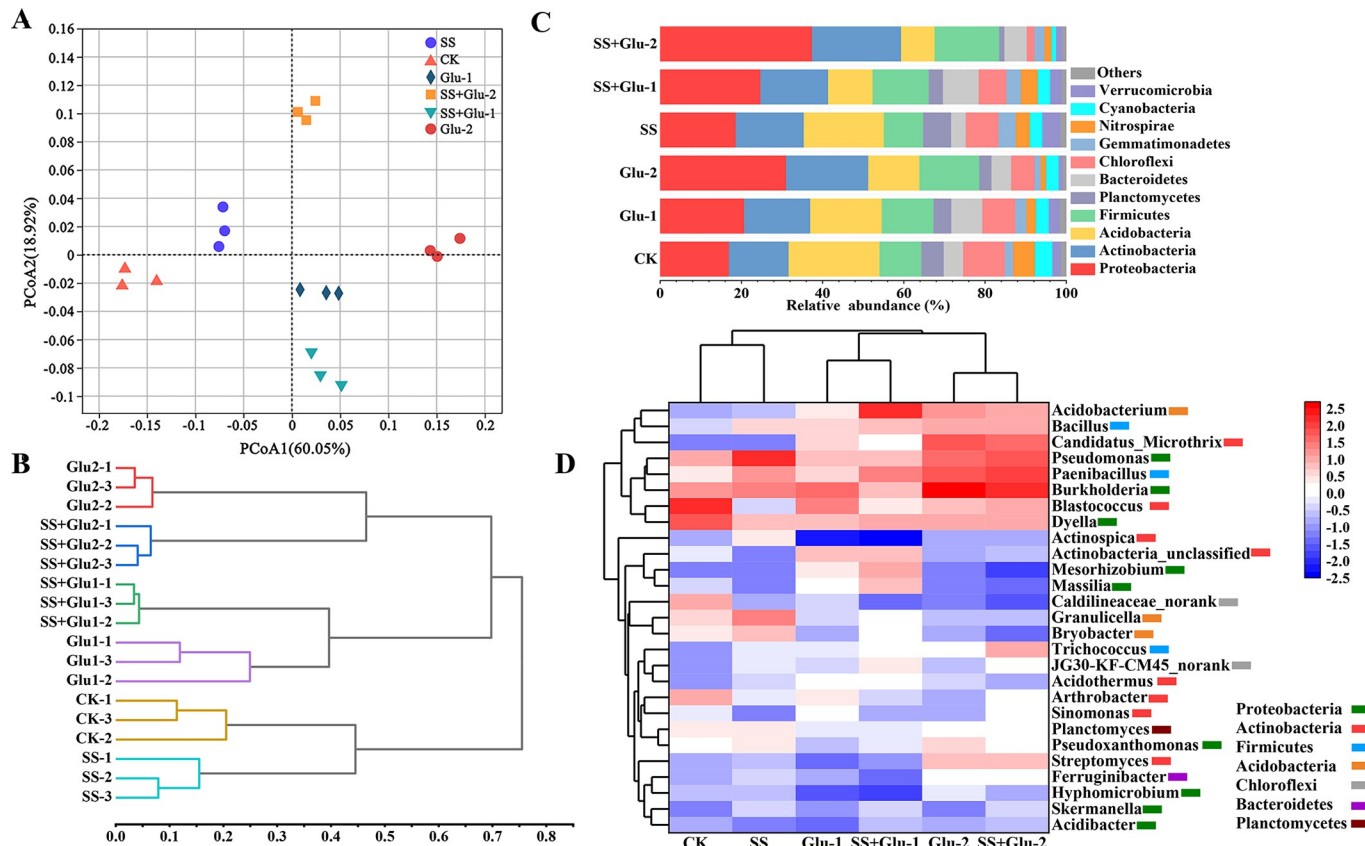

**Fig 3.** Principal coordinates analysis (PCoA, A), hierarchical cluster (B), the relative abundance of bacterial community composition at a phylum level (relative abundance exceeding to 1%, C), and heatmap of the relative abundance of bacterial genera (relative abundance exceeding to 1%, D) in the soils with glucose addition and sterilization at day 45. CK, non-sterilized soil without glucose addition; Glu-1, non-sterilized soil with low level of glucose addition; Glu-2, non-sterilized soil with high level of glucose addition; SS, sterilized soil without glucose addition; SS+Glu-1, sterilized soil with low level of glucose; SS+Glu-2, sterilized soil sterilization with high level of glucose addition.

*Acidibacter,*) *Firmicutes* (*Paenibacillus* and *Trichococcus*) and *Actinobacteria* (*Arthrobacter*, *Sinomonas* and *Blastococcus*) were higher, and those of *Proteobacteria* (*Mesorhizobium* and *Massilia*), *Chloroflexi* (*Caldilineaceae_norank*), *Acidobacteria* (*Bryobacter*) and *Actinobacteria* (*Acidothermus*) were lower in the SS+Glu-2 treatment relative to Glu-2 treatment. Moreover, the relative abundances of members of *Proteobacteria* (*Pseudomonas*) and *Firmicutes* (*Bacillus*) were higher, and those of *Actinobacteria* (*Blastococcus* and *Sinomonas*) and *Chloroflexi* (*Caldilineaceae_norank*) were lower in the SS treatment than those in the CK treatment.

## Root morphology and biomass

The root morphology indices (including surface area, volume, and length) were enhanced with the levels of glucose addition (Fig 4). Under the same level of glucose addition, root surface and volume were not significantly different ($P>0.05$) between sterilized soil and non-sterilized soil (Fig 4A and 4B). The SS+Glu-2 treatment significantly increased ($P<0.05$) the total root length by 5.9% compared with Glu-2 treatment (Fig 4C). The sterilized and non-sterilized soils with glucose addition increased the root biomass, on average, by 23.2% and 25.4% compared with those without glucose addition, respectively (Fig 4D).

## $NO_3^-$-N, $NO_2^-$-N and $NH_4^+$-N contents of root

The glucose addition significantly enhanced the $NO_3^-$N and $NO_2^-$N contents of root by 23.8% and 24.1%, but significantly decreased ($P<0.05$) the $NH_4^+$-N content of root by 11.7% (S5 Fig). The $NO_3^-$N content of root was significantly enhanced ($P<0.05$) by 13.1% and 16.5% in Glu-2 and SS+Glu-2 treatments compared with Glu-1 and SS+Glu-1 treatments, respectively (S5A Fig). The $NO_2^-$N content of root was significantly increased by 14.8% and 36.1% in the treatments of SS+Glu-1 and SS+Glu-2 compared with SS treatment, respectively (S5B Fig). The $NH_4^+$-N content of root in the Glu-2 and SS+Glu-2 treatments was 13.8% and 19.1% lower than that in the CK treatment, respectively (S5C Fig).

## Activities of enzymes involved in N metabolism of root

The activities of NR, GS, GOGAT and GPT were not significantly different ($P>0.05$) between Glu-1 and SS+Glu-1 treatments (Fig 5). The high level of glucose addition enhanced the NR activities by 21.9% and 28.6% in the sterilized and non-sterilized soils, respectively (Fig 5A). The Glu-2 and SS+Glu-2 treatments significantly increased the GS activities by 27.1% and 29.6% compared with the Glu-1 and SS+Glu-1 treatments, respectively (Fig 5B). The NADH-GDH activities in the SS treatment were higher than those in the CK treatment (Fig 5C). The GOGAT activities in the SS+Glu-1 and SS+Glu-2 treatments were 14.1% and 30.9% higher than those in the SS treatment (Fig 5D). Additionally, the GOT and GPT activities in the SS treatment were 23.3% and 10.3% higher than those in CK treatment, respectively (Fig 5E and 5F).

## Free amino acid contents of root

The contents of 16 free amino acids were increased with the levels of glucose addition (S2 Table). The most abundant amino acid content in roots was aspartic acid. The SS+Glu-2 and Glu-2 treatments significantly enhanced the contents of all amino acids by 2.2%~29.7% and by 5.1%~37.3% compared with SS and CK treatments, respectively. However, there was little variation ($P>0.05$) in the contents of lysine, glycine, serine, arginine, proline, and cysteine acids between SS and CK treatments. The SS+Glu-2 treatment considerably enhanced ($P<0.05$) the contents of aspartic and tyrosine acids by 3.9% and 9.1% compared with Glu-2 treatment, respectively.

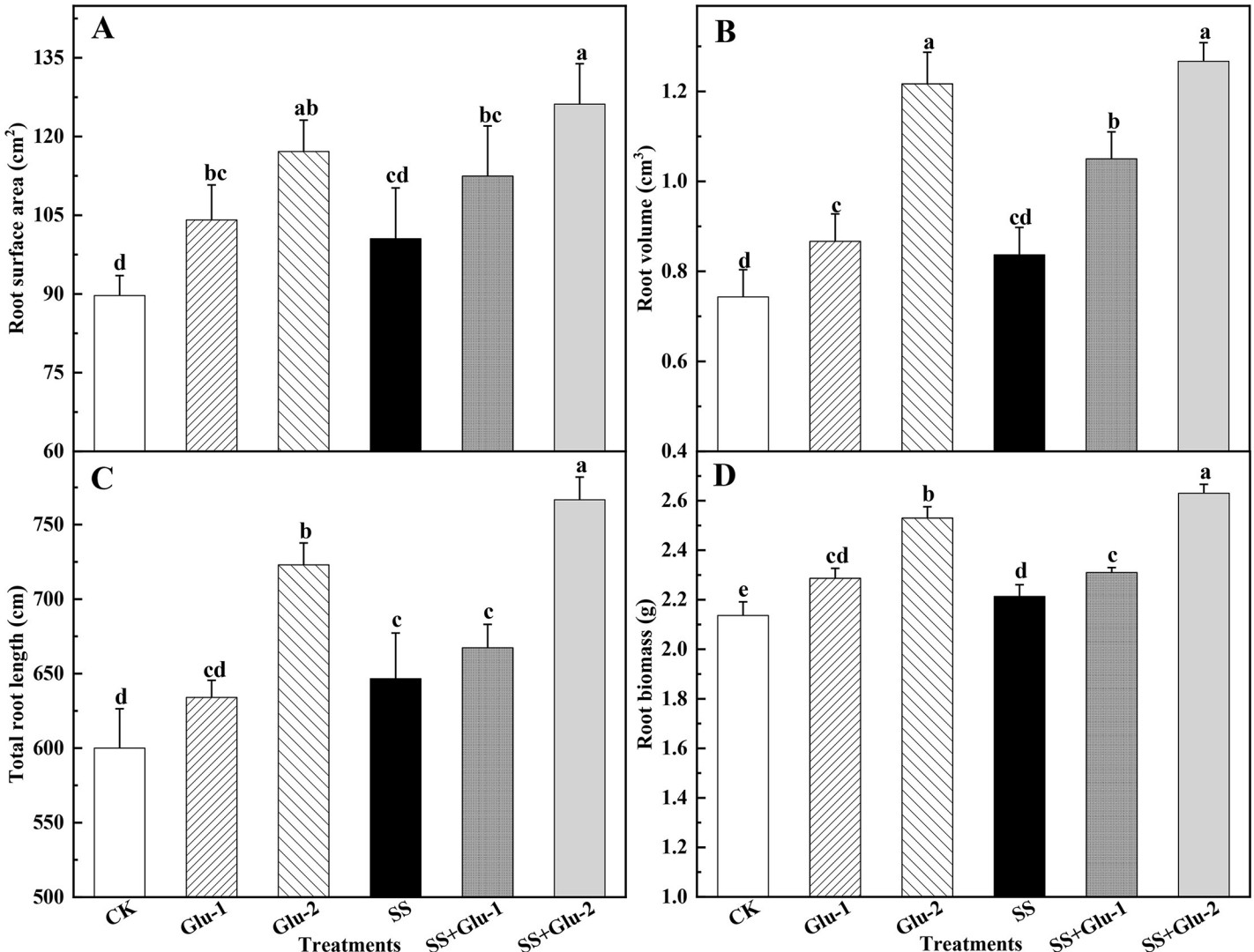

**Fig 4.** Root surface area (A), root volume (B), total root length (C) and root biomass (D) in the sterilized and non-sterilized soils with glucose addition at day 45. Different lowercase letters indicate significant differences between treatments ($P < 0.05$). CK, non-sterilized soil without glucose addition; Glu-1, non-sterilized soil with low level of glucose addition; Glu-2, non-sterilized soil with high level of glucose addition; SS, sterilized soil without glucose addition; SS+Glu-1, sterilized soil with low level of glucose; SS+Glu-2, sterilized soil sterilization with high level of glucose addition.

### Transcript levels of genes involved in N metabolism of root

Compared with SS treatment, the expression level of *NR* was increased by 1.5-fold and 2.8-fold in the SS+Glu-1 and SS+Glu-2 treatments, respectively (Fig 6A). The *GS* expression level was markedly higher ($P<0.05$) in the SS+Glu-1 and SS+Glu-2 treatments than that in the SS treatment (Fig 6B). The glucose addition increased the *GDH* mRNA levels, on average, by 4.4-fold and 2.5-fold in the non-sterilized and sterilized soils compared with no glucose addition, respectively (Fig 6C). The *GOGAT* transcript level was 1.5-fold higher in the sterilized soils than that in the non-sterilized soils (Fig 6D).

### Correlation between nitrogen metabolism of root and root morphology indices

The activities of root N metabolism enzyme were positively correlated with root surface ($R^2 = 0.832$, $P < 0.01$), root volume ($R^2 = 0.553$, $P < 0.05$) and root length ($R^2 = 0.756$, $P < 0.01$) as

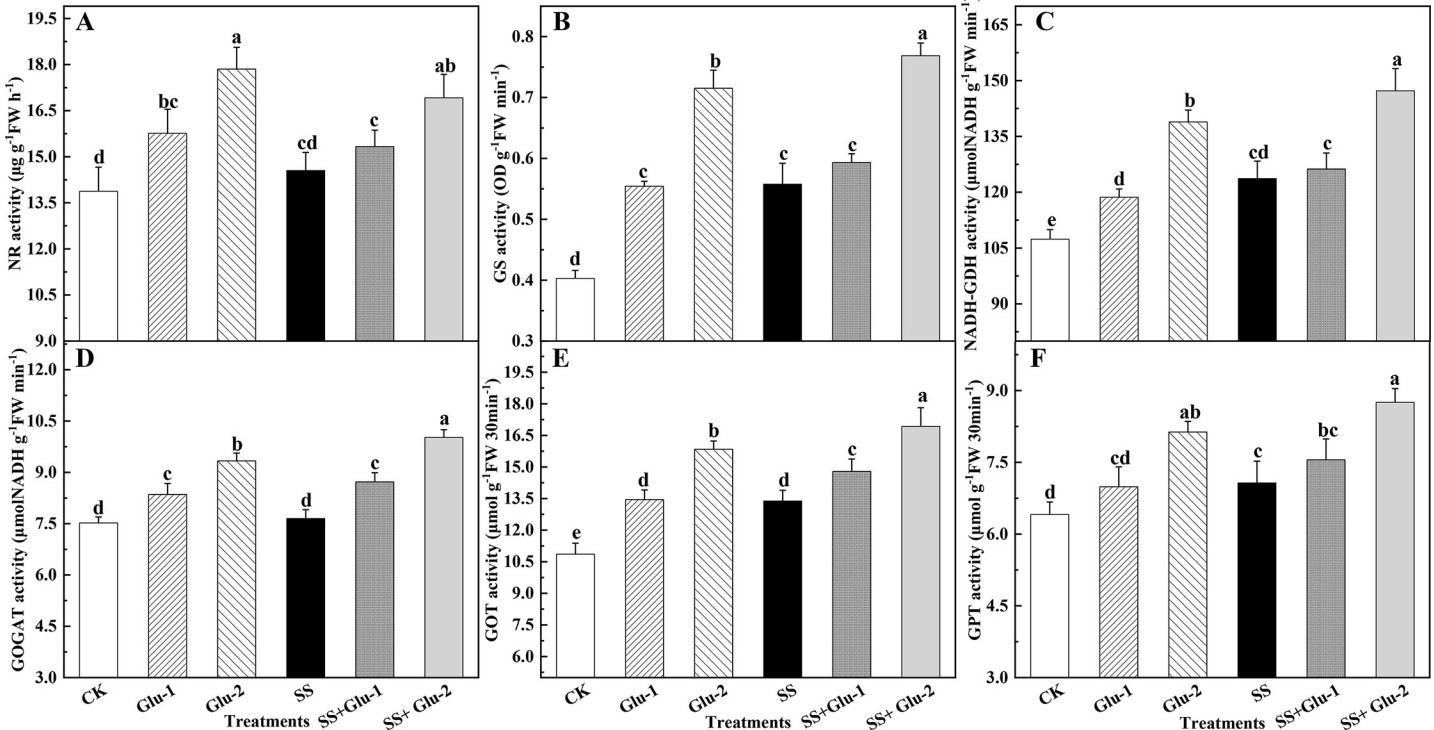

**Fig 5. Key enzyme activities involved in nitrogen metabolism of root in the sterilized and non-sterilized soils with glucose addition at day 45.** Different lowercase letters indicate significant differences between treatments ($P < 0.05$). CK, non-sterilized soil without glucose addition; Glu-1, non-sterilized soil with low level of glucose addition; Glu-2, non-sterilized soil with high level of glucose addition; SS, sterilized soil without glucose addition; SS+Glu-1, sterilized soil with low level of glucose; SS +Glu-2, sterilized soil sterilization with high level of glucose addition.

well as root biomass ($R^2 = 0.808$, $P < 0.01$), as Fig 7. This implies that N uptake and activities of metabolism enzyme plays an essential role in improving root morphology after glucose addition.

## Correlation among soil bacterial communities, soil organic carbon fractions and root morphology

The first two axes explained 80.43% and 11.41% of total variations, respectively (Fig 8). The first component separated the soils with glucose additions from the soils without glucose addition and explained 80.43% of total variation. MBC, TN, SOC, POC, *Proteobacteria*, *Actinobacteria* and *Firmicutes* were closely related. TN, MBC and WSOC significantly affected the morphology indices and $NO_3^-$N content of root. *Proteobacteria* had strong positive correlations with root length and biomass, but had strong negative correlations with $NH_4^+$-N content of root. Moreover, the $NO_3^-$N and $NO_2^-$N of root were positively correlated with SOC, TN and LOC fractions and negatively correlated with C/N ratio of soil.

## Discussion

### Dynamics of glucose C fixation and bacterial community in soils with sterilization and glucose addition

Input of exogenous C increased the SOC content [67]. However, some researches demonstrated that the total SOC content is not significantly increased even under excessive exogenous C input into soil [68,69]. The inconsistent results could be mainly because the trade-off between exogenous C and soil fertility together regulates the C sequestration in soil [70]. The

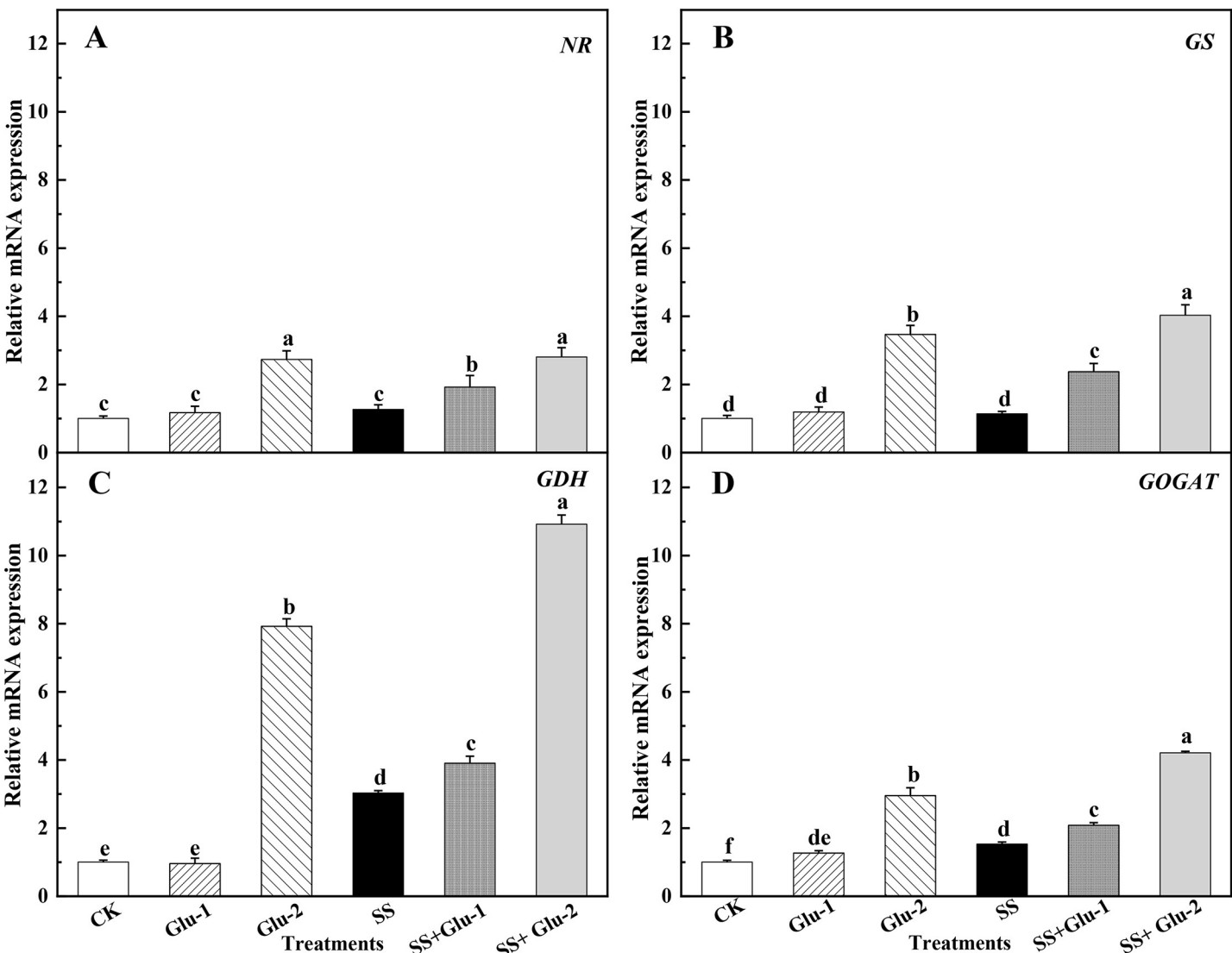

**Fig 6.** The transcript levels of genes involved in N metabolism of root in the sterilized and non-sterilized soils with glucose addition at day 45. *NR* (A), *GS* (B), *GDH* (C) and *GOGAT* (D). Different lowercase letters indicate significant differences between treatments ($P < 0.05$). CK, non-sterilized soil without glucose addition; Glu-1, non-sterilized soil with low level of glucose addition; Glu-2, non-sterilized soil with high level of glucose addition; SS, sterilized soil without glucose addition; SS+Glu-1, sterilized soil with low level of glucose; SS+Glu-2, sterilized soil sterilization with high level of glucose addition.

C fixed efficiency is lower in soils with high SOC than that in soils with low SOC content [71]. This could suggest that SOC content tends to be saturated with exogenous C addition, and the larger proportion of exogenous C remains in C-poor soil compared with C-rich soil closed to C saturation [72]. In our research, the soils with low nutrients availability could have the large potential of exogenous C fixation and promote SOC sequestration, which supported by net SOC balance (Fig 2).

The net SOC balance depends on the difference between new C gain and native SOC loss [73]. Glucose addition at low level leads to native SOC loss approximately equal to SOC gain in non-sterilized soil (Fig 2). Thus, the balance between accumulation and loss of SOC is probably associated with priming effect [21]. Glucose input to soil could activate dormant microorganisms, causing SOM decomposition [74]. But, new SOC formation derived from glucose C

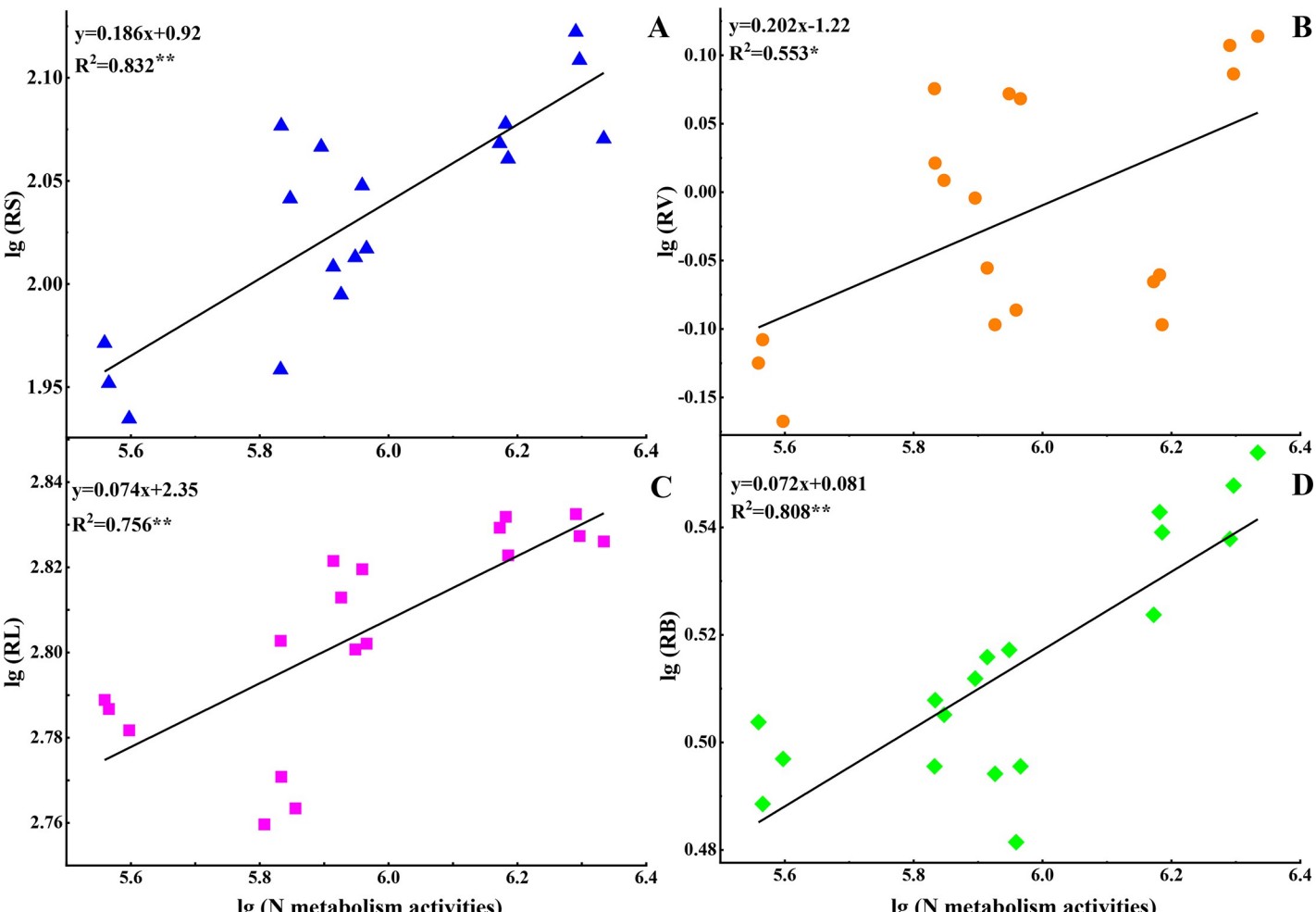

**Fig 7. Correlations between root nitrogen metabolism and root morphology indices in soils with glucose addition and sterilization at day 45** [**]. means $P < 0.01$, [*] means $P < 0.05$.

could offset native SOC loss in the sterilized soil with low level of glucose addition. Moreover, the residual rate of glucose-C in the sterilized soil was higher than that in non-sterilized soil under the same level of glucose addition (Fig 1). These results demonstrated that soil sterilization may promote glucose-C sequestration in a low C soil. The process of autoclaving could destroy soil structure and lead to soil particulate finer [75,76], which enhances available surface area and provide many sorption sites for glucose-C retention in soil [77].

Glucose input to soil supply energy source for microorganisms, and they are activated. WSOC as available C is assimilated by soil microorganisms, thereby accelerating the transformation of SOC in the soil [78]. Hence, the WSOC content was increased initially and then decreased with sampling time (S1C Fig). Exogenous glucose C is glued with soil clay particles [79,80], which could contribute to the formation of POC (S3D Fig). After glucose addition to soil, the POC content was increased with sampling time due to soil aggregate formation [81]. The formation of macroaggregate is enhanced with the levels of C input [82], while the decomposition of microaggregates rapidly induce the release of POC from aggregate occlusion and the increase of POC content. Our research did not explore the accumulation dynamics of POC in soil aggregates.

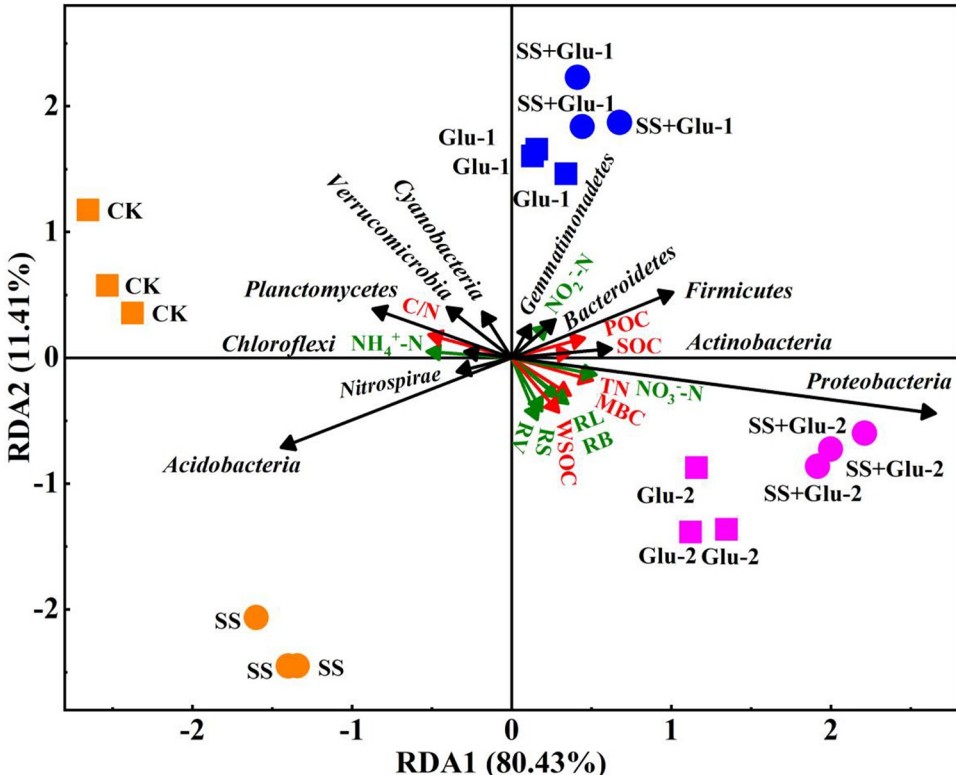

**Fig 8. Redundancy analysis of the relationship among soil bacterial communities (black arrow) at the phylum level, soil nitrogen and organic carbon fractions (red arrow), and root morphology indices (green arrow).** MBC, microbial biomass carbon; SOC, soil organic carbon; TN, total nitrogen; C/N, ratio of soil organic carbon to total nitrogen; POC, particulate organic carbon; WSOC, water-soluble organic carbon; $NH_4^+$-N, ammonium nitrogen of root; $NO_3^-$N, nitrate nitrogen of root; $NO_2^-$N, nitrite nitrogen of root; RV, root volume; RS, root surface; RL, root length; RB, root biomass. Square and circle denote the non-sterilized and sterilized soils, respectively. CK, non-sterilized soil without glucose addition; Glu-1, non-sterilized soil with low level of glucose addition; Glu-2, non-sterilized soil with high level of glucose addition; SS, sterilized soil without glucose addition; SS+Glu-1, sterilized soil with low level of glucose; SS+Glu-2, sterilized soil sterilization with high level of glucose addition.

The glucose addition increased the richness and alpha-diversities of soil bacteria (Table 1). Similar results were found in the rhizosphere of *Cerasus sachalinensis* Kom [83]. The glucose addition enhanced the relative abundances of specific bacterial communities (including *Proteobacteria*, *Actinobacteria* and *Firmicutes* at the phylum level) in the low C soil, which is C-limited for microbial growth relative to N nutrient. Both *Proteobacteria* and *Actinobacteria* as copiotrophic bacteria (*r*-strategists) have strong abilities to utilize labile organic C source [84,85]. While the relative abundance of *Proteobacteria* exceeded to that of *Actinobacteria* at day 45. *Actinobacteria* taxa own few amounts of high affinity transporters to transport specific substrate, which leads to their saturated proliferation under C-poor condition. Additionally, *Actinobacteria* may strongly compete soil nutrients with *Proteobacteria* [86]. A negative correlation between *Acidobacteria* and total SOC was found (Fig 8), which could be attributed that high level of glucose addition may produce disorder osmotic and aberrant growth of *Acidobacteria* cells as oligotrophic bacteria [87,88]. The increase in soil TN content drives the shift of dominant microbial growth strategies from *K*-to *r*-strategists [89,90]. Soils with higher level of glucose addition had higher TN content and lower C/N ratio (S2 Fig). And C/N was negatively associated with *Proteobacteria*, *Actinobacteria* and *Firmicutes* (opportunistic bacteria, *r*-

strategists) (Fig 8). These results demonstrated that *r*-strategy decomposers rather than *K*-strategists dominate at lower substrate C/N ratios.

The composition of bacterial communities at the phylum level was similar, while their relative abundances were different between sterilized and non-sterilized soils. Autoclave sterilization for short term (4 h) kills most of native soil microorganisms, leaving many empty niches for recolonized microbe to fill. On the other hand, plant growth could favor the recolonization of highly desirable microorganisms, which are used to assimilate root exudates as "food source", once the niches competition was removed via sterilization [42,91]. Therefore, microorganism occupied empty niches in the sterilized soils planted with apple seedlings. *Pseudomonas*, preferring root exudate C to the other C source, is beneficial bacteria for most plants [92]. This bacteria genus could firstly occupy empty niches in the sterilized soil, which may be the reason that *Pseudomonas* was more enriched in the SS treatment than those in the CK treatment. Regardless of sterilization or not, soil bacterial communities in the treatments without glucose addition were well separated from those in the treatments with glucose addition along the first component, which explained 60.05% of total variation (Fig 3A). This result indicated that the available C substrate supply plays dominate roles in bacterial communitie variation of low C soil. Moreover, the relative abundance of *Acidobacterium*, belonged to *Acidobacteria* (*K*-strategists), was higher than that *Bacillus*, belonged to *Firmicutes* (*r*-strategists), in the soil combined with sterilization and low level of glucose addition (Fig 3D). The *K*-strategists always dominates under low nutrients availability conditions, but are consistent with their outcompeting *r*-strategists when resources are limited [93,94].

### Root morphology varied with exogenous C addition and soil sterilization

Plant roots grow in the soil, exploring soil nutrients and water availability [95]. Soil bacterial community structure has substantial influences on the growth and health of plant by regulating root morphology and development [96]. The glucose addition and/or soil sterilization significantly increased the indices of root morphology. This was mainly associated with the increase in the relative abundance of *Burkholderia* (*Proteobacteria*), *Paenibacillus* (*Firmicutes*) and *Streptomyces* (*Actinobacteria*) by glucose addition and/or soil sterilization. The bacterial phyla of *Proteobacteria*, which consists mostly of G⁻ bacteria and diazotrophs, classified as plant growth promoting rhizobacteria (PGPR) [97,98]. This PGPR exerts a beneficial effect on root growth on the production of phytohormone (e.g. IAA) or on nutrients uptake (e.g. N) by plants [99]. Some the other genera, such as *Streptomyces*, also have PGPR traits to hydrolyse chitin [100]. The genus *Paenibacillus* is easily isolated from the agricultural soil and rhizosphere [101] and has the characterise against pathogens [102]. Therefore, the above mentioned genera improved root morphology and promoted plant growth due to root hormones levels producing IAA in the rhizosphere and the nutrients availability. Moreover, the growth of rhizosheaths is regulated by the cooperation of soil particles with root hair and polysaccharides released either by root exudates or colonized bacteria. While the larger amount of soluble C release, in sterilized and glucose addition soils (S1C Fig), favours the growth of exopolysaccharides-producing bacteria [39], which could cause more soil aggregates attached on root surface and extend root length. In addition, glucose addition could improve soil stoichiometric ratio of C, N and P and create a suitable environment for soil microbial growth, which in turn could favour root growth in low C soil [103].

### Enzymes activities involved in N metabolism of root respond to glucose C addition and soil sterilization

Optimal root zone environment is beneficial to orderly root metabolism process [104]. Our results showed that glucose addition and soil sterilization increased the organic acid content of

*M. baccata* root (S3 Table). The addition of glucose, regardless of soil sterilization, provides available C source for soil microorganism and increases microbial activities [105], which could promote the energy metabolism of root (S6 Fig), root activity (S7 Fig), as well as nutrient absorption by root.

N is an essential mineral nutrient that has profound impacts on plant growth and crop yield. N absorbed by roots is mainly derived from fertiliser, necromass decomposition and native soil organic matter mineralisation. $NO_3^-$N is the main N source for root. Glucose addition and/or soil sterilization increased the root length and volume (Fig 4), which could enhance the potential ability of N absorption by root. This also could be supported by the highly positive correlation between root morphology and enzymes activities of N metabolism (Fig 7).

Additionally, the exogenous C addition changes the ratio of SOC to TN, and further causes the competition between soil microorganism and plant root for soil N nutrient [106]. All the assimilation of $NO_3^-$N and $NO_2^-$N are reduced to $NH_4^+$-N through NR [107]. However, the $NH_4^+$-N content of root under glucose addition and/or soil sterilization was significantly decreased (S5C Fig). Root assimilates $NH_4^+$-N and rapidly converts it into organic compounds through GS, NADH-GOGAT and NADH-GDH [108]. On the whole, glucose addition and/or soil sterilization enhanced the absorption and assimilation of N in root due to the activities of NR, GS, NADH-GDH, NADH-GOGAT (Fig 5) and the mRNA genes expression level (Fig 6), which, in turn, decreased the $NH_4^+$-N content in root. $NH_4^+$-N as an amino donor of proline is converted to glutamate through NADH-GDH activities and mRNA genes expression, which were higher in SS+Glu-2 treatment than that in the other treatments. This was consistent with the increase of glutamic and proline contents in root (S2 Table). Moreover, glutamate can be further converted to aspartic acid or alanine by GOT or GPT, respectively. Our results obtained that the increased activities of GOT and GPT in high level of glucose addition appeared high content of aspartic or alanine (Fig 5E and 5F). Hence, the supply of available C source for low C soil promotes the exogenous C fixation in soil, optimizes root zone environment, improves root morphology, and further enhances the key enzymic activities of N metabolism and mRNA expression in apple roots.

## Conclusions

Glucose addition combined with soil sterilization not only increased the SOC content and new SOC formation derived from glucose C, but also increased the alpha diversity and changed bacterial community structure in soils. Although soil microbial communities were similar between non-sterilization and sterilization, soil sterilization mainly increased the relative abundances of *Proteobacteria*, *Firmicutes* and *Verrucomicrobia* at the phyla level. Furthermore, the glucose addition, especially combined with soil sterilization improved root morphology, promoted the potential abilities of root N metabolism, and increased the amino acid synthesis in root. Overall, these results suggested the supply of C substrate with healthy soil conditions well shapes soil microbial communities and root morphology, and potentially increases soil C sequestration in agroecosystems. However, the complexity of C substrate drives the function and structure of soil microbial communities, which could lead to dynamics of plant growth and soil nutrient transformation. Further research should be focused on the coupled mechanism among nutrients transformation, plant growth and soil C sequestration under supply of complicated C substrates for low C soil.

## Supporting information

**S1 Fig.** Contents of total soil organic carbon (A), microbial biomass carbon (B), water soluble organic carbon (C) and particulate organic carbon (D) in the sterilized and non-sterilized soils

with glucose addition at day 45. Different uppercase letters indicate significant differences ($P < 0.05$) among different treatments at the same sampling time. Different lowercase letters indicate significant differences ($P < 0.05$) among different sampling time within the same treatment. Overlapping date points with the same significant differences are indicated by common letters. CK, non-sterilized soil without glucose addition; Glu-1, non-sterilized soil with low level of glucose addition; Glu-2, non-sterilized soil with high level of glucose addition; SS, sterilized soil without glucose addition; SS+Glu-1, sterilized soil with low level of glucose addition; SS+Glu-2, sterilized soil with high level of glucose addition.
(DOCX)

**S2 Fig.** Content of total nitrogen (TN) of soil (A) and ratio of total soil organic carbon to TN (C/N, B) in the sterilized and non-sterilized soils with glucose addition at day 45. CK, non-sterilized soil without glucose addition; Glu-1, non-sterilized soil with low level of glucose addition; Glu-2, non-sterilized soil with high level of glucose addition; SS, sterilized soil without glucose addition; SS+Glu-1, sterilized soil with low level of glucose addition; SS+Glu-2, sterilized soil with high level of glucose addition. Different lowercase letters indicate significant differences ($P < 0.05$) among treatments.
(DOCX)

**S3 Fig.** $\delta^{13}C$ values of soil organic carbon (A), microbial biomass carbon (B), water soluble organic carbon (C) and particulate organic carbon (D) in the sterilized and non-sterilized soils with glucose addition. Different uppercase letters indicate significant differences ($P < 0.05$) among different treatments at the same sampling time. Different lowercase letters indicate significant differences ($P < 0.05$) among different sampling time within the same treatment. Overlapping date points with the same significant differences are indicated by common letters. CK, non-sterilized soil without glucose addition; Glu-1, non-sterilized soil with low level of glucose addition; Glu-2, non-sterilized soil with high level of glucose addition; SS, sterilized soil without glucose addition; SS+Glu-1, sterilized soil with low level of glucose addition; SS +Glu-2, sterilized soil with high level of glucose addition.
(DOCX)

**S4 Fig. Heatmap of the abundant bacterial family (relative abundance exceeding to 1%) in the sterilized and non-sterilized soils with glucose addition at day 45.** CK, non-sterilized soil without glucose addition; Glu-1, non-sterilized soil with low level of glucose addition; Glu-2, non-sterilized soil with high level of glucose addition; SS, sterilized soil without glucose addition; SS+Glu-1, sterilized soil with low level of glucose addition; SS+Glu-2, sterilized soil with high level of glucose addition.
(DOCX)

**S5 Fig.** Contents of $NO_3^-N$ (A), $NO_2^-N$ (B) and $NH_4^+$-N (C) of apple root in the sterilized and non-sterilized soils with glucose addition at day 45. Different lowercase letters indicate significant differences between treatments ($P < 0.05$). CK, non-sterilized soil without glucose addition; Glu-1, non-sterilized soil with low level of glucose addition; Glu-2, non-sterilized soil with high level of glucose addition; SS, sterilized soil without glucose addition; SS+Glu-1, sterilized soil with low level of glucose addition; SS+Glu-2, sterilized soil with high level of glucose addition.
(DOCX)

**S6 Fig. Enzymes activities related to energy metabolism at day 45.** PEPC (A, phosphoenolpyruvate carboxylase), MDH (B, malate dehydrogenase) and ICDH (C, isocitrate dehydrogenase). Different lowercase letters indicate significant differences between treatments

($P < 0.05$). CK, non-sterilized soil without glucose addition; Glu-1, non-sterilized soil with low level of glucose addition; Glu-2, non-sterilized soil with high level of glucose addition; SS, sterilized soil without glucose addition; SS+Glu-1, sterilized soil with low level of glucose addition; SS+Glu-2, sterilized soil with high level of glucose addition.
(DOCX)

**S7 Fig. Root vitality of *Malus baccata* (L.) Borkh. in the sterilized and non-sterilized soils with glucose addition at day 45.** Different lowercase letters indicate significant differences between treatments ($P < 0.05$). CK, non-sterilized soil without glucose addition; Glu-1, non-sterilized soil with low level of glucose addition; Glu-2, non-sterilized soil with high level of glucose addition; SS, sterilized soil without glucose addition; SS+Glu-1, sterilized soil with low level of glucose addition; SS+Glu-2, sterilized soil with high level of glucose addition.
(DOCX)

**S1 Table. Gene-specific primers used for quantitative real-time PCR in the sterilized and non-sterilized soils with glucose addition.** CK, non-sterilized soil without glucose addition; Glu-1, non-sterilized soil with low level of glucose addition; Glu-2, non-sterilized soil with high level of glucose addition; SS, sterilized soil without glucose addition; SS+Glu-1, sterilized soil with low level of glucose addition; SS+Glu-2, sterilized soil with high level of glucose addition.
(DOCX)

**S2 Table. Change in amino acid contents of root in the sterilized and non-sterilized soils with glucose addition at day 45.** Different lowercase letters indicate significant differences between treatments ($P < 0.05$). CK, non-sterilized soil without glucose addition; Glu-1, non-sterilized soil with low level of glucose addition; Glu-2, non-sterilized soil with high level of glucose addition; SS, sterilized soil without glucose addition; SS+Glu-1, sterilized soil with low level of glucose addition; SS+Glu-2, sterilized soil with high level of glucose addition.
(DOCX)

**S3 Table. Change in organic acid contents of root in the sterilized and non-sterilized soils with glucose addition at day 45.** Different lowercase letters indicate significant differences between treatments ($P < 0.05$). CK, non-sterilized soil without glucose addition; Glu-1, non-sterilized soil with low level of glucose addition; Glu-2, non-sterilized soil with high level of glucose addition; SS, sterilized soil without glucose addition; SS+Glu-1, sterilized soil with low level of glucose addition; SS+Glu-2, sterilized soil with high level of glucose addition.
(DOCX)

## Acknowledgments

We deeply appreciate Dr. Tingting An (College of Land and Environment, Shenyang Agricultural University, China) for her assistance in this study. Special thanks to the editors and the anonymous reviewers for their thorough and constructive comments.

## Author Contributions

**Data curation:** Bianbin Qi.

**Funding acquisition:** Sijun Qin, Deguo Lyu.

**Investigation:** Bianbin Qi, Kuo Zhang.

**Supervision:** Sijun Qin, Deguo Lyu, Jiali He.

**Writing – original draft:** Bianbin Qi.

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
