## [Decision Letter · Decision Letter 0]

28 Aug 2021

PONE-D-21-22234

Glucose addition promotes C fixation and bacteria community composition in C-poor
soils, improves root morphology, and enhances key N metabolism in apple roots

PLOS ONE

Dear Dr. Qin,

Thank you for submitting your manuscript to PLOS ONE. After careful consideration, we
feel that it has merit but does not fully meet PLOS ONE’s publication criteria as it
currently stands. Therefore, we invite you to submit a revised version of the
manuscript that addresses the points raised during the review process.

Please submit your revised manuscript by Oct 11 2021 11:59PM. If you will need more
time than this to complete your revisions, please reply to this message or contact
the journal office at plosone@plos.org. When
you're ready to submit your revision, log on to https://www.editorialmanager.com/pone/ and select the 'Submissions
Needing Revision' folder to locate your manuscript file.

If you would like to make changes to your financial disclosure, please include your
updated statement in your cover letter. Guidelines for resubmitting your figure
files are available below the reviewer comments at the end of this letter.

We look forward to receiving your revised manuscript.

Kind regards,

Ying Ma, Ph.D.

Academic Editor

PLOS ONE

Journal Requirements:

5. Please upload a new copy of Figure 5, 7 and 8 as the detail is not clear. Please
follow the link for more information: https://blogs.plos.org/plos/2019/06/looking-good-tips-for-creating-your-plos-figures-graphics/"
https://blogs.plos.org/plos/2019/06/looking-good-tips-for-creating-your-plos-figures-graphics/

Reviewers' comments:

Reviewer's Responses to Questions

**Comments to the Author**

1. Is the manuscript technically sound, and do the data support the conclusions?

Reviewer #1: Yes

Reviewer #2: Partly

2. Has the statistical analysis been performed
appropriately and rigorously? 

Reviewer #1: Yes

Reviewer #2: Yes

3. Have the authors made all data underlying the
findings in their manuscript fully available?

Reviewer #1: No

Reviewer #2: No

4. Is the manuscript presented in an intelligible
fashion and written in standard English?

Reviewer #1: Yes

Reviewer #2: No

5. Review Comments to the Author

Reviewer #1: Reviewers' Comments to Authors: the manuscript presents the potential to
be published in PLOS ONE Journal. Please consider the questions raised below.

Abstract p.2 l.23: Present the meaning of the acronym SOC the first time it appears
in the text.

Abbreviations p.3 l.42: Insert meaning of acronyms when they first appear in the
text.

Materials and Methods p.9 l.193: Explain better the steps of DNA extraction,
sequencing, and data analysis (How did you check the quality of the extracted DNA?
Where and how was the Illumina sequencing performed? What are the steps of data
analysis using bioinformatics?).

Materials and Methods p.9 l.193: You must submit the raw FASTQ files from the
sequencing to NCBI and inform the study accession number.

Materials and Methods p.9 l.197: Separate "Miseqmachine".

Materials and Methods p.10 l.202: Insert the term "analytical balance".

Result p.12 l.251: Replace for "Results".

Result p.14 l.319: I didn't find Good coverage results (Good, 1953). I suggest
presenting these data, which demonstrate the quality of the sequencing.

Result p.14 l.319: Why has soil bacterial diversity not been analyzed? (beta and
alpha diversity?). These analyzes contribute to the understanding of the structural
dynamics of the community after the application of treatments.

Result p.14 l.319: I raise a reflection on relative abundance results: There is an
ongoing debate about the downside of interpreting sequencing data based on its
relative abundance (which can cause data distortions). When we convert data into
relative abundances, the independent variable appears to be correlated, since the
frequency of any microbial community must add 1. To solve this problem, it is more
feasible to use mathematical approaches for data composition analysis. The most
accepted approach so far is Aitchison's centered logarithmic ratio transformation
(clr).

Read:

Gloor GB, Reid G. Compositional analysis: a valid approach to analyze microbiome
high-throughput sequencing data. Can J Microbiol. 2016;62(8):692-703.
doi:10.1139/cjm-2015-0821

Aitchison J. The Statistical Analysis of Compositional Data. Journal of the Royal
Statistical Society Series B (Methodological). 1982;44(4):38.

Result p.14 l.319: A differential abundance analysis would also be interesting to
highlight soil bacterial genera affected by the treatments.

Result p.14 l.319: Improve the description of phyla and genera found in each
treatment (text is confusing).

Result p.14 l.339: In general, clearly demonstrate which treatment presented the best
results (text is confusing).

Discussion p.22 l.430: Explore further the effect of soil sterilization on your
bacterial community structure beyond the addition of glucose.

Discussion p.22 l.430: The contribution of soil bacteria to apple root growth
remained to be discussed more. It would be interesting to correlate changes in
community structure with candidate taxa for growth promotion.

Conclusion p.25 l.527: Better explain the applicability/projections of this study to
field conditions.

Reviewer #2: The paper entitled “Glucose addition promotes C fixation and bacteria
community composition in C-poor soils, improves root morphology, and enhances key N
metabolism in apple roots” reports an interesting approach where the authors added
glucose to the sterilized and non-sterilized soil. Then, they studied the response
of the plant, soil C fractionation, and their bacterial composition. To publish in
Plos One journal is necessary improve the following:

-The manuscript careful language revision and standardize the use of American or
British English.

-The authors should extend the discussion, including the dynamic of r and k
microorganisms when glucose is added.

-The results section is long. I considered that some results can be in the
supplementary material.

-Discuss the effect of C addition on C:N ratio and which is their relationship with
the results observed.

-The manuscript should show what is the importance of these results to
agriculture

Specific comments:

Title:

-The term bacterial community is don't used correctly. Did the Glucose addition
promote the bacterial community?

Abstract

- Please describe the abbreviation before the first citation. For example, “SOC”.

Line 30-31: The authors should check the use of the term "richness". Did they
calculate the richness indexes?

Keywords:

-Avoid the use of abbreviations, for example: C and N.

Introduction:

-Line 74:78: For me is not clear, how in soil with N depletion the addition of C can
increase the N absorption by the plant?

-Line 79-80: Here the authors should approach the dynamic of r and k microorganisms
depending of the C source complexity added?

-Line 83: Replace the term “biological desert”

Materials and methods:

-Line 125: Please, clarify the term “debris”.

-Line 127-129: Describe the references used to perform the chemical characterization
of the soil.

-Line 145-146. The authors should have irrigated the two soils (sterilized and
unsterilized) with sterile water.

-Line 146: change “injetected” for “applied”.

-Line 150-153: The authors should clearly describe the soil collection conditions.
Was the soil influenced by the roots?

-Line 155: Clarify the statistical designs. The authors use 30 seedling roots per
replicate or treatment.

-Line 166: change “rpm” for “xg”

-Line 193: The sequencing data should be deposited in a database.

-Line 193: Change “microbial” for “bacterial”

-In this section the authors should give more details of methodology used to analyze
the data (bioinformatic analyses). How was performed the DNA extraction? for
sequencing, did you use kit V2 or V3?

-Line 196: The authors should include the reference of primers used.

-Line 204: The authors should include the reference of technique used.

-Line 230-231: Please check the redaction.

Results:

-In the figures and tables presented, the authors should include the abbreviation
definitions in legends.

-Line 255-258: Why does the glucose addition increases the SOC only in low doses?

-Line 320: Please, analyze the data to family level.

-Line 326-327: My suggestion is to compare with the data of soil after
sterilization.

-Line 340: Is curiously that root biomass increased in C.

Fig 10. What is the meaning the color in the correlation test.

Discussion

-Line 442-444: This paragraph is not clear, please rewriting.

-Line 444: The authors should not affirm that the glucose increased the “net C
sequestration”.

-Line 477-479: How the addition and sterilization increase the indices of root
morphology. I consider this result contradictory.

-Line 490-491: The authors should cite the similar studies mentioned.

Conclusions

- The data presented does not permit conclude on changes in bacterial richness when
applied glucose or/and sterilize the soil.

6. PLOS authors have the option to publish the peer
review history of their article (what does this mean?). If published, this will
include your full peer review and any attached files.

If you choose “no”, your identity will remain anonymous but your review may still be
made public.

**Do you want your identity to be public for this peer review?** For
information about this choice, including consent withdrawal, please see our
Privacy Policy.

Reviewer #1: No

Reviewer #2: No

---

## [Author Response · Author response to Decision Letter 0]

23 Oct 2021

Responses to Comments

Reviewer #1: 

The manuscript presents the potential to be published in PLOS ONE Journal. Please
consider the questions raised below.

Abstract p.2 l.23: Present the meaning of the acronym SOC the first time it appears
in the text.

Answer: We changed “SOC” to “soil organic C (SOC)” in Line 23 because the acronym
appeared the first time in the manuscript.

Abbreviations p.3 l.42: Insert meaning of acronyms when they first appear in the
text.

Answer: We substituted “Low C soil” to “Low carbon soil”, “Exogenous C” to “Exogenous
carbon”, and “N metabolism” to “Nitrogen metabolism” in Line 41. 

Materials and Methods p.9 l.193: Explain better the steps of DNA extraction,
sequencing, and data analysis (How did you check the quality of the extracted DNA?
Where and how was the Illumina sequencing performed? What are the steps of data
analysis using bioinformatics?).

Answer: The detailed steps of DNA extraction, sequencing, and data analysis were
shown as follows:

Genomic DNA was extracted from soil samples at 45-day using a FastDNA Spin Kit (Omega
Bio-Tek, Norcross, GA, USA) according to the manufacturer’s protocol. The quality of
DNA was analysed with 1% agarose gel electrophoresis and the total quantity of DNA
was determined using a Thermo NanoDrop 2000 UV Microvolume Spectrophotometer (Thermo
Fisher Scientific, USA). The primers 338F (5'-ACTCCTACGGGAGGCAGCAG-3') and 806R
(5'-GGACTACHVGGGTWTCTAAT-3') were chosen to amplify the 16S rRNA genes in the V3-V4
regions [49]. The PCR amplification conditions included an initial denaturation at
95 ℃ for 3 min, followed by 27 cycles of denaturation at 95 ℃ for 30 s, annealing at
60 ℃ for 30 s, extension at 72 °C for 30 s, and a finial extension at 72 ℃ for 10
min. The PCR products of all samples were purified with a Cycle Pure Kit (OMEGA),
pooled in equimolar concentrations and performed on an Illumina (2 � 300
bp) MiSeq machine (Illumina, San Diego, CA, USA) at the Shanghai Origingene
Biotechnology Co. Ltd., China.

The paired-end reads were analysed statistically by Trimmomatic software after
depletion of primers. Bases of reads with a tail mass of 20 bp or less, overlapping
paired-end reads less than 10 bp, and box sequences at both ends of reads were
filtered. The unmatched sequences and singletons were excluded according to the
Silva reference database v128 [50]. The operational taxonomic units were defined by
clustering nonrepetitive sequences at 97% similarity and classified according to the
Silva reference database using the Ribosomal Database Project Bayesian algorithm
classifier (RDP) [51]. Then, Usearch version 7.1 was used to cluster the sequences
with 97% similarity for operational taxonomic units (OTU) [52]. 

Difference in the composition of bacterial OTUs according to taxonomic category
between treatments was assessed. After centred-log ratio (clr) transformation (log
transformation of the geometric mean), the ‘codaSeq.clr’ function was used in the
‘CoDaSeq’ package of R software [53]. The alpha diversity indices of bacterial
communities, including ACE, Shannon and Simpson, were analysed using ‘phyloseq’
package of R software. Principal coordinate analysis (PCoA) and redundancy analysis
(RDA) were performed using ‘stats’ and ‘vegan’ packages in R software [54],
respectively.

We also added the detailed steps of DNA extraction, sequencing, and data analysis in
Line 203-229 of the Materials and Methods Section. We also added the new references
in the “References” section. The cited references in Materials and Methods were
shown as follows: 

49. Fadrosh DW, Ma B, Gajer P, Sengamalay N, Ott S, Ravel J, et al. An improved
dual-indexing approach for multiplexed 16s rRNA gene sequencing on the Illumina
Miseq platform. Microbiome. 2014;2:1-7. doi: 10.1186/2049-2618-2-6

50. Quast C, Pruesse E, Yilmaz P, Gerken J, Schweer T, Yarza P, et al. The SILVA
ribosomal RNA gene database project: improved data processing and web-based tools.
Nucleic Acids Res. 2012;41(D1):D590-D596. doi: 10.1093/nar/gks1219

51. Wang Q, Garrity GM, Tiedje JM, Cole JR. Naive Bayesian classifier for rapid
assignment of rRNA sequences into the new bacterial taxonomy. Appl. Environ.
Microbiol. 2007;73(16):5261-5267. doi: org/10.1128/AEM.00062-07

52. Edgar RC. Search and clustering orders of magnitude faster than BLAST.
Bioinformatics, 2010;26(19):2460-2461. doi.org/10.1093/bioinformatics/btq461

53. Gloor GB and Reid G. Compositional analysis: a valid approach to analyze
microbiome high-throughput sequencing data. Can. J. Microbiol. 2016;62(8):692-703.
doi.org/10.1139/cjm-2015-0821

54. Dixon P. VEGAN, a package of R functions for community ecology. J. Veg. Sci.
2003, 14(6): 927-930. doi.org/10.1111/j.1654-1103.2003.tb02228.x

Materials and Methods p.9 l.193: You must submit the raw FASTQ files from the
sequencing to NCBI and inform the study accession number.

Answer: The raw FASTQ files were deposited in the National Center for Biotechnology
Information (NCBI) and the Sequence Read Archive (SRA) number was PRJNA765206.
Please see Line 230.

Materials and Methods p.9 l.197: Separate "Miseqmachine".

Answer: We changed “Miseqmachine” to “Miseq machine” in Line 213.

Materials and Methods p.10 l.202: Insert the term "analytical balance".

Answer: We changed “The root was oven-dried for 24 h at 80 ℃ for dry weight analysis”
into “The root was oven-dried for 24 h at 80 ℃, and weighed with electronic
analytical balance” in Line 236.

Result p.12 l.251: Replace for "Results".

Answer: We changed “Result” to “Results” in Line 278. Thanks for your careful
review.

Result p.14 l.319: I didn't find Good coverage results (Good, 1953). I suggest
presenting these data, which demonstrate the quality of the sequencing.

Answer: Thanks for your professional suggestion. We added the coverage data in Table
1 so as to demonstrate the quality of the sequencing. The coverage exceeded 0.994 in
all treatments. Please see Line 342 and Table 1. 

Result p.14 l.319: Why has soil bacterial diversity not been analyzed? (beta and
alpha diversity?). These analyzes contribute to the understanding of the structural
dynamics of the community after the application of treatments.

Answer: We analyzed soil bacterial alpha diversity indices (including Chao, ACE,
Shannon, Simpson) and beta diversity in order to understand the contribution of soil
sterilization and glucose addition to the structural dynamics of the community. The
values of Ace, Chao and Shannon indices were increased with the glucose addition
levels, whereas the Simpson index showed the opposite trend. Compared with
non-sterilized soil, sterilized soil with high level of glucose addition increased
the values of Ace, Chao and Shannon indices by 13.8%, 7.9% and 11.4%, respectively;
and sterilized soil with low level of glucose addition increased them by 17.3%,
13.8% and 2.6%, respectively; The PCoA plot based on the clr-transformed data was
used to presented the changes of bacterial community structures in soil. The first
two principal components explained 78.9% of total variations in the composition of
bacterial communities (Fig 3A). The PCoA1 clearly separated the treatments with and
without glucose addition. The non-sterilized and sterilized treatments were
differentiated along the PCoA2. As presented by the hierarchical cluster analysis,
bacterial communities revealed two clusters comprising samples from all treatment
groups (Fig 3B). The treatments of non-sterilized soil and sterilized soil with the
same level of glucose addition clustered together.

We added these data about alpha and beta diversity in Table 1 and Figure 3 and these
results in Line 342-392. 

We also added the discussion on the contribution of soil sterilization and glucose
addition to soil bacterial communities in Line 526-561. Thanks for your professional
comments.

Result p.14 l.319: I raise a reflection on relative abundance results: There is an
ongoing debate about the downside of interpreting sequencing data based on its
relative abundance (which can cause data distortions). When we convert data into
relative abundances, the independent variable appears to be correlated, since the
frequency of any microbial community must add 1. To solve this problem, it is more
feasible to use mathematical approaches for data composition analysis. The most
accepted approach so far is Aitchison's centered logarithmic ratio transformation
(clr). 

Answer: Thanks for your professional suggestion. We totally agreed with you. We
log-transformed the data of soil bacterial communities. After centred-log ratio
(clr) transformation (log transformation of the geometric mean), the ‘codaSeq.clr’
function was used in the ‘CoDaSeq’ package of R software. Please see Line
223-229.

Result p.14 l.319: A differential abundance analysis would also be interesting to
highlight soil bacterial genera affected by the treatments.

Answer: We added the difference analysis of relative abundance of bacterial community
at the genera level among treatments. “The heatmap of soil bacterial genera showed
that all samples were clustered into two groups consisting of the treatment with
glucose addition and that without glucose addition (Fig 3D). The relative abundances
of members of Proteobacteria (Pseudomonas, Skermanella and Acidibacter,) Firmicutes
(Paenibacillus and Trichococcus) and Actinobacteria (Arthrobacter, Sinomonas and
Blastococcus) were higher, and those of Proteobacteria (Mesorhizobium and Massilia),
Chloroflexi (Caldilineaceae_norank), Acidobacteria (Bryobacter) and Actinobacteria
(Acidothermus) were lower in the SS+Glu-2 treatment relative to Glu-2 treatment.
Moreover, the relative abundances of members of Proteobacteria (Pseudomonas) and
Firmicutes (Bacillus) were higher, and those of Actinobacteria (Blastococcus and
Sinomonas) and Chloroflexi (Caldilineaceae_norank) were lower in the SS treatment
than those in the CK treatment.” Please see Line 383-392.

Result p.14 l.319: Improve the description of phyla and genera found in each
treatment (text is confusing).

Answer: We rewrote the description about the difference analysis of relative
abundances of bacterial communities at the phylum and genera levels among
treatments. Please see Line 375-380 and Line 383-392. The detailed revision was
shown as follows: The predominant bacterial phyla in all treatments were
Proteobacteria, Actinobacteria, and Acidobacteria, with relative abundances larger
than 10% (Fig 3C). The glucose addition enhanced the relative abundances of
Proteobacteria, Actinobacteria and Firmicutes by 59.4%, 19.6% and 43.3%, but
decreased those of Acidobacteria and Chloroflexi by 41.8% and 39.1%, respectively.
The relative abundances of Proteobacteria, Actinobacteria and Verrucomicrobia were
higher in the SS treatment than those in the CK treatment.

Result p.14 l.339: In general, clearly demonstrate which treatment presented the best
results (text is confusing).

Answer: We rewrote the results about root morphology and biomass in order to make
them clear. Please see Line 394-400. The detailed revision was showed as
follows:

The root morphology indices (including surface area, volume, and length) were
enhanced with the levels of glucose addition (Fig 4). Under the same level of
glucose addition, root surface and volume were not significantly different
(P>0.05) between sterilized soil and non-sterilized soil (Figs 4A and 4B). The
SS+Glu-2 treatment significantly increased (P<0.05) the total root length by 5.9%
compared with Glu-2 treatment (Fig 4C). The sterilized and non-sterilized soils with
glucose addition increased the root biomass, on average, by 23.2% and 25.4% compared
with those without glucose addition, respectively (Fig 4D). 

Discussion p.22 l.430: Explore further the effect of soil sterilization on your
bacterial community structure beyond the addition of glucose.

Answer: We further discussed the effect of soil sterilization on bacterial community
structure in Line 544-561 of the Discussion section. “The composition of bacterial
communities at the phylum level was similar, while their relative abundances were
different between sterilized and non-sterilized soils. Autoclave sterilization for
short term (4 h) kills most of native soil microorganisms, leaving many empty niches
for recolonized microbe to fill. On the other hand, plant growth could favor the
recolonization of highly desirable microorganisms, which are used to assimilate root
exudates as “food source”, once the niches competition was removed via sterilization
[42, 91]. Therefore, microorganism occupied empty niches in the sterilized soils
planted with apple seedlings. Pseudomonas, preferring root exudate C to the other C
source, is beneficial bacteria for most plants [92]. This bacteria genus could
firstly occupy empty niches in the sterilized soil, which may be the reason that
Pseudomonas was more enriched in the SS treatment than those in the CK treatment.
Regardless of sterilization or not, soil bacterial communities in the treatments
without glucose addition were well separated from those in the treatments with
glucose addition along the first component, which explained 60.05% of total
variation (Fig 3A). This result indicated that the available C substrate supply
plays dominate roles in bacterial communitie variation of low C soil. Moreover, the
relative abundance of Acidobacterium, belonged to Acidobacteria (K-strategists), was
higher than that Bacillus, belonged to Firmicutes (r-strategists), in the soil
combined with sterilization and low level of glucose addition (Fig 3D). The
K-strategists always dominates under low nutrients availability conditions, but are
consistent with their outcompeting r-strategists when resources are limited [93,
94].” 

Discussion p.22 l.430: The contribution of soil bacteria to apple root growth
remained to be discussed more. It would be interesting to correlate changes in
community structure with candidate taxa for growth promotion.

Answer: We further discussed the contribution of soil bacteria to apple root growth
in Line 566-577. “The glucose addition and/or soil sterilization significantly
increased the indices of root morphology. This was mainly associated with the
increase in the relative abundance of Burkholderia (Proteobacteria), Paenibacillus
(Firmicutes) and Streptomyces (Actinobacteria) by glucose addition and/or soil
sterilization. The bacterial phyla of Proteobacteria, which consists mostly of G-
bacteria and diazotrophs, classified as plant growth promoting rhizobacteria (PGPR)
[97, 98]. This PGPR exerts a beneficial effect on root growth on the production of
phytohormone (e.g. IAA) or on nutrients uptake (e.g. N) by plants [99]. Some the
other genera, such as Streptomyces, also have PGPR traits to hydrolyse chitin [100].
The genus Paenibacillus is easily isolated from the agricultural soil and
rhizosphere [101] and has the characterise against pathogens [102]. Therefore, the
above mentioned genera improved root morphology and promoted plant growth due to
root hormones levels producing IAA in the rhizosphere and the nutrients
availability.” 

Conclusion p.25 l.527: Better explain the applicability/projections of this study to
field conditions.

Answer: Thanks for your valuable suggestions. Now we have the highlight the intention
and modified the conclusion (lines 616-628) in the following sentences: “Glucose
addition combined with soil sterilization not only increased the SOC content and new
SOC formation derived from glucose C, but also increased the alpha and beta
diversities of soil bacterial communities. Although soil microbial communities were
similar between non-sterilization and sterilization, soil sterilization mainly
increased the relative abundances of Proteobacteria, Firmicutes and Verrucomicrobia
at the phyla level. Furthermore, the glucose addition, especially combined with soil
sterilization improved root morphology, promoted the potential abilities of root N
metabolism, and increased the amino acid synthesis in root. Overall, these results
suggested the supply of C substrate with heathy soil conditions well shapes soil
microbial communities and root morphology, and potentially increases soil C
sequestration in agroecosystems. However, the complexity of C substrate drives the
function and structure of soil microbial communities, which could lead to dynamics
of plant growth and soil nutrient transformation. Further research should be focused
on the coupled mechanism among nutrients transformation, plant growth and soil C
sequestration under supply of complicated C substrates for low C soil.”

Reviewer: 2

The paper entitled “Glucose addition promotes C fixation and bacteria community
composition in C-poor soils, improves root morphology, and enhances key N metabolism
in apple roots” reports an interesting approach where the authors added glucose to
the sterilized and non-sterilized soil. Then, they studied the response of the
plant, soil C fractionation, and their bacterial composition. To publish in Plos One
journal is necessary improve the following:

The manuscript careful language revision and standardize the use of American or
British English.

Answer: We made careful revision of English language. Dr. Tingting An visited the
University of Tennessee for more than one years as a scholar, and published more
than 7 SCI papers as the first author or corresponding author. Dr. An made
substantial revisions to the manuscript in order to make it more readable. We hope
that the revised manuscript is readable and meets the standards for publication.

The authors should extend the discussion, including the dynamic of r and k
microorganisms when glucose is added.

Answer: We further discussed the dynamics of r and k microorganisms after the
addition of glucose to soil. The glucose addition enhanced the relative abundances
of specific bacterial communities (including Proteobacteria, Actinobacteria and
Firmicutes at the phylum level) in the low C soil, which is C-limited for microbial
growth relative to N nutrient. Both Proteobacteria and Actinobacteria as
copiotrophic bacteria (r-strategists) have strong abilities to utilize labile
organic C source [84, 85]. While the relative abundance of Proteobacteria exceeded
to that of Actinobacteria at day 45. Actinobacteria taxa own few amounts of high
affinity transporters to transport specific substrate, which leads to their
saturated proliferation under C-poor condition. Additionally, Actinobacteria may
strongly compete soil nutrients with Proteobacteria [86]. A negative correlation
between Acidobacteria and total SOC was found (Fig 8), which could be attributed
that high level of glucose addition may produce disorder osmotic and aberrant growth
of Acidobacteria cells as oligotrophic bacteria [87, 88]. Please see
Line527-538.

The results section is long. I considered that some results can be in the
supplementary material.

Answer: Thanks for your valuable comments. We listed some results as supplementary
material. Please see the file of supplementary material.

Discuss the effect of C addition on C:N ratio and which is their relationship with
the results observed.

Answer: We further discussed the the effect of C addition on C:N ratio and which is
their relationship with the results observed. “The increase in soil TN content
drives the shift of dominant microbial growth strategies from K-to r-strategists
[89, 90]. Soils with higher level of glucose addition had higher TN content and
lower C/N ratio (S2 Fig). And C/N was negatively associated with Proteobacteria,
Actinobacteria and Firmicutes (opportunistic bacteria, r-strategists) (Fig 8). These
results demonstrated that r-strategy decomposers rather than K-strategists dominate
at lower substrate C/N ratios.” Please see Line538-543.

The manuscript should show what is the importance of these results to
agriculture.

Answer: Now we have the highlight the intention and modified the conclusion (lines
616-628) in the following sentences:

“Glucose addition combined with soil sterilization not only increased the SOC content
and new SOC formation derived from glucose C, but also increased the alpha and beta
diversities of soil bacterial communities. Although soil microbial communities were
similar between non-sterilization and sterilization, soil sterilization mainly
increased the relative abundances of Proteobacteria, Firmicutes and Verrucomicrobia
at the phyla level. Furthermore, the glucose addition, especially combined with soil
sterilization improved root morphology, promoted the potential abilities of root N
metabolism, and increased the amino acid synthesis in root. Overall, these results
suggested the supply of C substrate with heathy soil conditions well shapes soil
microbial communities and root morphology, and potentially increases soil C
sequestration in agroecosystems. However, the complexity of C substrate drives the
function and structure of soil microbial communities, which could lead to dynamics
of plant growth and soil nutrient transformation. Further research should be focused
on the coupled mechanism among nutrients transformation, plant growth and soil C
sequestration under supply of complicated C substrates for low C soil.”

Specific comments:

Title: The term bacterial community is don't used correctly. Did the Glucose addition
promote the bacterial community?

Answer: We totally agreed with you that “bacterial community” is not suitably used.
We added the data and results about soil bacterial diversity (beta and alpha
diversity) in Lines 342-355, Table 1 found that glucose addition markedly increased
the bacterial community diversities. We changed the title into “Glucose addition
promotes C fixation and bacteria diversity in C-poor soils, improves root
morphology, and enhances key N metabolism in apple roots”. Thanks for your valuable
comments.

Abstract: Please describe the abbreviation before the first citation. For example,
“SOC”. 

Answer: We changed “SOC” into “soil organic C (SOC)”. Please see Line 24. 

Abstract: Line 30-31: The authors should check the use of the term "richness". Did
they calculate the richness indexes?

Answer: Thanks for your professional comment. We added the values of Chao, Ace, and
Shannon indices to describe bacteria community richness and diversity in Line
342-348 and Table 1 of the Results section. We found that the glucose addition
increased the richness and diversity indices of soil bacterial community compared
with no-glucose addition. We changed “Bacterial community richness” to “Bacterial
community richness and diversity” in Line 31. 

Keywords: Avoid the use of abbreviations, for example: C and N.

Answer: We substituted the abbreviation of C and N with “carbon” and “nitrogen” in
the Keywords, in Line 41.

Introduction: Line 74-78: For me is not clear, how in soil with N depletion the
addition of C can increase the N absorption by the plant?

Answer: The “nitrogen mining and competition” theory (presented by Fontaine et al.,
2011) interprets that easily available C addition stimulates soil microbial growth
in the rhizosphere, leading to the mining of additional N from soil organic matter
(SOM) by soil microorganisms, that is, the production of some extracellular enzymes
and thus enhancement of subsequent SOM decomposition (Fontaine et al., 2003). Living
microorganisms require soil nutrients for their growths. Hence, the competition
between plants and microorganisms mainly for N and phosphorus (P) in
nutrient-limited soils (Kirkby et al., 2011). Microorganisms, as high surface area
to volume ratios, show substantially faster initial uptake of all N forms (Fischer
et al., 2010). However, the short life cycle of rhizosphere microbes and
unidirectional N flux from soil to roots facilitates the transform of N from
microorganisms to roots (Rosswall T, 1982; Schmidt et al., 2007). On the other hand,
the addition of easily available C is depleted within a few days via microbial
utilization and decomposition. The imbalance between absence of new C input and
continuous consumption of old C by soil microorganisms leads to the release of N
from microbial necromass into the soil, which results in the availability of N for
plants (Yakov and Xu, 2013). 

The following references have been used to support above explanation:

Fontaine S, Mariotti A, Abbadie L. The priming effect of organic matter: a question
of microbial competition? Soil Biology and Biochemistry, 2003, 35(6): 837-843. doi:
org/10.1016/S0038-0717(03)00123-8

Kirkby CA, Kirkegaard JA, Richardson AE, et al. Stable soil organic matter: a
comparison of C:N:P:S ratios in Australian and other world soils. Geoderma, 2011,
163(3-4): 197-208. doi: org/10.1016/j.geoderma.2011.04.010

Fischer H, Ingwersen J, Kuzyakov Y. Microbial uptake of low‐molecular‐weight organic
substances out‐competes sorption in soil. European Journal of Soil Science, 2010,
61(4): 504-513. doi: org/10.1111/j.1365-2389.2010.01244.x

Rosswall T. Microbiological regulation of the biogeochemical nitrogen cycle. Plant
and Soil, 1982, 67(1): 15-34. doi: 10.1007/978-94-009-7639-9_2

Schmidt SK, Costello EK, Nemergut DR, et al. Biogeochemical consequences of rapid
microbial turnover and seasonal succession in soil. Ecology, 2007, 88(6): 1379-1385.
doi: org/10.1890/06-0164

Schimel JP, Weintraub MN. The implications of exoenzyme activity on microbial carbon
and nitrogen limitation in soil: a theoretical model. Soil Biology and Biochemistry,
2003, 35(4): 549-563. doi: org/10.1016/S0038-0717(03)00015-4 

Kuzyakov Y, Xu X. Competition between roots and microorganisms for nitrogen:
mechanisms and ecological relevance. New Phytologist, 2013, 198(3): 656-669. doi:
10.1111/nph.12235

Hence, we have added more elaboration on the relationship between exogenous C
addition and N absorption by root in introduction. We rewrote this paragraph and
focused on microbial strategies responded to C substrate and soil nutrients in order
to make it clear. Please see Line 74-83.

Introduction: Line 79-80: Here the authors should approach the dynamic of r and k
microorganisms depending of the C source complexity added?

Answer: We add some comments on the dynamics of r and K microorganisms affected by
the exogenous C source complexity. Please see Line 74-80.

“Moreover, relative abundances of microorganism with different growth strategies are
varied with soil nutrient environments, which would shape different soil microbial
community structure [18]. In general, K-strategists have lower growth rates and
higher substrate affinities. Conversely, r-strategists have higher growth rates,
lower substrate affinities and preferentially assimilate labile C [19]. The supply
of labile C leads to the succession of microorganism from r- to K-strategists, and
this process mainly depends on nitrogen (N) captured by soil microorganisms
[20].”

Introduction: Line 83: Replace the term “biological desert”.

Answer: We changed “biological desert” to “leaves numerous empty niches for
microorganism re-colonization” in order to make the sentence clear. Please see Line
87.

Materials and methods: Line 125: Please, clarify the term “debris”.

Answer: We firstly picked out the visible plant roots, rock pieces and the other
debris from soil samples before experiment. We changed this sentence into “We picked
out the visible plant root, rock pieces and the other debris from soil samples,
passed them through a 2 mm sieve, and then fully mixed for pot experiments.” in
order to make this sentence clear. Please see Line 130-131.

Materials and methods: Line 127-129: Describe the references used to perform the
chemical characterization of the soil.

Answer: Thanks for your careful reading. We add the references used to perform soil
basic properties in Line 135-138. The measured methods of SOC, total N, δ13C value,
and MBC were showed in the following section, and the contents of available N,
available P, available K, and pH value were analysed with the methods by Le and
Marschner [37], and particle size separation was carried out with the method by
Jensen et al. [38].

Materials and methods: Line 145-146. The authors should have irrigated the two soils
(sterilized and unsterilized) with sterile water.

Answer: We really irrigated unsterilized soils with tap water so as to stimulate
plant growth condition in fields (Qin et al.2014; Li et al. 2019; Moreira et al.
2019). We would irrigate unsterilized soil with sterile water to avoid the effect of
irrigated water on soil microbial community in the future research. We added the
cited references in Line 155.

Qin SJ, Zhou WJ, D Lyu, Liu LZ. Effects of soil sterilization and biological agent
inoculation on the root respiratory metabolism and plant growth of Cerasus
sachalinensis Kom. Sci. Hortic. 2014;170(1):189-195. doi:
10.1016/j.scienta.2014.03.019

Li K, DiLegge M J, Minas I S, et al. Soil sterilization leads to re-colonization of a
healthier rhizosphere microbiome. Rhizosphere. 2019;12:100176. doi:
10.1016/j.rhisph.2019;100176

Moreira H, Pereira S I A, Marques A P G C, et al. Effects of soil sterilization and
metal spiking in plant growth promoting rhizobacteria selection for phytotechnology
purposes. Geoderma. 2019;334:72-81. doi: 10.1016/j.geoderma.2018.07.025

Materials and methods: Line 146: change “injetected” for “applied”

Answer: We changed “injetected” to “applied” in Line 154.

Materials and methods: Line 150-153: The authors should clearly describe the soil
collection conditions. Was the soil influenced by the roots?

Answer: After glucose addition for 3, 7, 15, 30, and 45 days, soil samples were
randomly collected from five pots with the similar seedling growth (one pot as one
replication) per treatment. The aboveground seedings were firstly cut at the root
base, and then the roots and soil cores remained in the pots were destructively
collected. The soil samples adhered to root were carefully separated with shaking
method because the seeding roots occupied the whole pots. After being removed the
visible roots, the collected soil sub-samples were mixed thoroughly, and then were
divided into half for further analysis.

The soil was influenced by the roots because the seedling root occupied the whole pot
(internal diameter 10 cm, height 12 cm). We added the detailed conditions for soil
sample collection. Please see Line 157-162. 

Materials and methods: Line 155: Clarify the statistical designs. The authors use 30
seedling roots per replicate or treatment.

Answer: Thanks for your careful revision. We collected 5 pots (1 seedling per pot)
per treatment, and set up 6 treatments. Hence, we collected 30 seedling roots (5
seedlings/treatment � 6 treatments) in total on day 45. Sorry for our
miswriting.

Materials and methods: Line 166: change “rpm” for “� g”

Answer: We changed 4000 rpm to 3000 � g in Line 175.

Materials and methods: Line 193: The sequencing data should be deposited in a
database.

Answer: The raw FASTQ files were deposited in the National Center for Biotechnology
Information (NCBI) and Sequence Read Archive (SRA) number was PRJNA765206. We added
this information in Line 230-231. Thanks for your professional comments.

Materials and methods: Line 193: Change “microbial” for “bacterial.

Answer: We changed “Soil microbial community” to “DNA extraction, PCR amplification
and bioinformatic analysis of bacteria” to make the sentence clear. Please see Line
202. 

Materials and methods: Line 193: In this section the authors should give more details
of methodology used to analyze the data (bioinformatic analyses). How was performed
the DNA extraction? for sequencing, did you use kit V2 or V3?

Answer: We added the more details on DNA extraction, sequencing, and data analysis in
Line 203-229. “Genomic DNA was extracted from soil samples at 45-day using a FastDNA
Spin Kit (Omega Bio-Tek, Norcross, GA, USA) according to the manufacturer’s
protocol. The quality of DNA was analysed with 1% agarose gel electrophoresis and
the total quantity of DNA was determined using a Thermo NanoDrop 2000 UV Microvolume
Spectrophotometer (Thermo Fisher Scientific, USA). The primers 338F
(5'-ACTCCTACGGGAGGCAGCAG-3') and 806R (5'-GGACTACHVGGGTWTCTAAT-3') were chosen to
amplify the 16S rRNA genes in the V3-V4 regions [49]. The PCR amplification
conditions included an initial denaturation at 95 ℃ for 3 min, followed by 27 cycles
of denaturation at 95 ℃ for 30 s, annealing at 60 ℃ for 30 s, extension at 72 °C for
30 s, and a finial extension at 72 ℃ for 10 min. The PCR products of all samples
were purified with a Cycle Pure Kit (OMEGA), pooled in equimolar concentrations and
performed on an Illumina (2 � 300 bp) MiSeq machine (Illumina, San Diego,
CA, USA) at the Shanghai Origingene Biotechnology Co. Ltd., China.

The paired-end reads were analysed statistically by Trimmomatic software after
depletion of primers. Bases of reads with a tail mass of 20 bp or less, overlapping
paired-end reads less than 10 bp, and box sequences at both ends of reads were
filtered. The unmatched sequences and singletons were excluded according to the
Silva reference database v128 [50]. The operational taxonomic units were defined by
clustering nonrepetitive sequences at 97% similarity and classified according to the
Silva reference database using the Ribosomal Database Project Bayesian algorithm
classifier (RDP) [51]. Then, Usearch version 7.1 was used to cluster the sequences
with 97% similarity for operational taxonomic units (OTU) [52]. 

Difference in the composition of bacterial OTUs according to taxonomic category
between treatments was assessed. After centred-log ratio (clr) transformation (log
transformation of the geometric mean), the ‘codaSeq.clr’ function was used in the
‘CoDaSeq’ package of R software [53]. The alpha diversity indices of bacterial
communities, including ACE, Shannon and Simpson, were analysed using ‘phyloseq’
package of R software. Principal coordinate analysis (PCoA) and redundancy analysis
(RDA) were performed using ‘stats’ and ‘vegan’ packages in R software [54],
respectively.”

Materials and methods: Line 196: The authors should include the reference of primers
used.

Answer: Thanks for your thoughtful comments. We added the reference of primers used
in Line 209. The cited reference was as following:

Fadrosh DW, Ma B, Gajer P, Sengamalay N, Ott S, Ravel J, et al. An improved
dual-indexing approach for multiplexed 16s rRNA gene sequencing on the Illumina
Miseq platform. Microbiome. 2014; 2:1-7. doi: 10.1186/2049-2618-2-6

Materials and methods: Line 204: The authors should include the reference of
technique used.

Answer: We added the reference of technique used in Line 234. The cited reference was
as following:

Agapit C, Gigon A, Blouin M. Earthworm effect on root morphology in a split root
system. Plant Biosyst. 2018; 152(4):780-786. doi: doi.org/10.1080/11263504.2017.1338627

Materials and methods: Line 230-231: Please check the redaction. 

Answer: We revised the sentence into “About 0.2 g frozen root sample and 2 mL
reaction agent (50 mmol L-1 Tris-HCl with pH 8.0, 2 mmol L-1 MgCl2, 2 mmol L-1 DTT
and 0.4 mol L-1 sucrose) were fully homogenized” in order to make it clear. Please
see Lines 252-253.

Results: In the figures and tables presented, the authors should include the
abbreviation definitions in legends.

Answer: We added the abbreviation definitions in legends in all the figures and
tables in order to make them clear. Please see all the figures and tables.

Results: Line 255-258: Why does the glucose addition increases the SOC only in low
doses?

Answer: Exogenous C addition to sterilization soil can stimulate microorganisms
increase, thereby promoting the increase of microbial biomass (Esch, et al. 2013).
This increased microbial biomass can supply more microbial necromass for forming
recalcitrant C and benefit SOC accumulation (Schmidt et al., 2011). To a certain
extent, the quantity of activated microorganisms increased with the amount of labile
C added. In our study, sterilization soil results in clay particles gathered and
increase the surface and available sorption site to glucose-C. Hence, residual rate
of glucose-C with higher level glucose in SS+Glu-2 treatment increase than Glu-2.
Thereby actives more amounts of microorganisms and produces larger microbial biomass
and prominently contribute to sequestration of SOC. But the lower level of glucose
addition showed the opposite trend. Lower level of glucose-C addition resulted in
actived microbe produces less microbial biomass and contribute to less sequestration
of SOC in SS+Glu-1. 

The following references have been used to support above explanation:

Esch, E. H., Hernandez, D. L., Pasari, J. R., Kantor, R. S. G., & Selmants, P. C.
(2013). Response of soil microbial activity to grazing, nitrogen deposition, and
exotic cover in a serpentine grassland. Plant and Soil, 366(1–2), 671–682. https://doi.org/10.1007/s11104-012-1463-5

Schmidt, M. W., Torn, M. S., Abiven, S., Dittmar, T., Guggenberger, G., Janssens, I.
A., Trumbore, S. E. (2011). Persistence of soil organic matter as an ecosystem
property. Nature, 478, 49–56. https://doi.org/10.1038/nature10386

Results: Line 320: Please, analyze the data to family level.

Answer: We presented the data on soil microbial community structure at the family
level in S4 Fig and added the related results in Line 380-382. 

Results: Line 326-327: My suggestion is to compare with the data of soil after
sterilization.

Answer: We rewrote the results and focused on the comparison of data after
sterilization. Please see Line 383-392. The detailed revisions were showed as
follows:

The heatmap of soil bacterial genera showed that all samples were clustered into two
groups consisting of the treatment with glucose addition and that without glucose
addition (Fig 3D). The relative abundances of members of Proteobacteria
(Pseudomonas, Skermanella and Acidibacter,) Firmicutes (Paenibacillus and
Trichococcus) and Actinobacteria (Arthrobacter, Sinomonas and Blastococcus) were
higher, and those of Proteobacteria (Mesorhizobium and Massilia), Chloroflexi
(Caldilineaceae_norank), Acidobacteria (Bryobacter) and Actinobacteria
(Acidothermus) were lower in the SS+Glu-2 treatment relative to Glu-2 treatment.
Moreover, the relative abundances of members of Proteobacteria (Pseudomonas) and
Firmicutes (Bacillus) were higher, and those of Actinobacteria (Blastococcus and
Sinomonas) and Chloroflexi (Caldilineaceae_norank) were lower in the SS treatment
than those in the CK treatment.

Results: Line 340: Is curiously that root biomass increased in C.

Answer: Thanks for your valuable comments. Root biomass increased in SS treatment may
be attributed to the following reasons: (1) sterilization removes pathogen and
alleviates disease suppression (Sosnowski et al., 2009; Savory, 1966). (2) the high
temperature (121 ℃) and pressure (2 bar) during autoclaving affects the physical and
chemical structure of the soil and liberates labile C, such as WSOC, and N (Mahmood
et al., 2014; Berns et al., 2008), which enhances root development and acquisition
of nutrients, then increases the root biomass (3) Since the original soils had been
sterilized prior to reinoculation, the community assemblage recolonization phase was
primarily influenced by the new habitat and the adequacy of its conditions for each
of the inoculated taxon (Mallon et al., 2015), and increased availability of labile
C and N (due to steam sterilization) may have influenced the new community formation
by promoting the growth of copiotrophic microorganisms. Those copiotrophic
microorganisms classified as plant growth promoting rhizobacteria (PGPR) (Dommelen
et al, 2007; Ali et al, 2015), benefited root growth and nutrient absorption,
thereby improved the biomass of root. 

The following references have been cited to support above reasons:

Sosnowski MR, Fletcher JD, Daly AM, Rodoni BC, Viljanen-Rollinson SLH. Techniques for
the 

treatment, removal and disposal of host material during programmes for plant pathogen
eradication. Plant Pathol. 2009;58(4):621-635.
doi:10.1111/j.1365-3059.2009.02042.x

Savory BM. Specific replant diseases causing root necrosis and growth depression in
perennial fruit and plantation crops. Farnham Royal. 1966 (1).

Marschner B, Bredow A. Temperature effects on release and ecologically relevant
properties of dissolved organic carbon in sterilised and biologically active soil
samples. Soil Biol. Biochem. 2002;

34:459e466. doi: 10.1016/S0038-0717(01)00203-6

Mahmood T, Mehnaz S, Fleischmann F, Ali R, Hashmi ZH, Iqbal Z. Soil sterilization
effects on root growth and formation of rhizosheaths in wheat seedlings.
Pedobiologia. 2014;57:123-130. doi: org/10.1016/j.pedobi.2013.12.005

Berns AE, Philipp H, Narres HD, Burauel P, Vereecken H, Tappe W. Effect of
gamma-sterilization and autoclaving on soil organic matter structure as studied by
solid state NMR, UV and fluorescence spectroscopy. Eur. J. Soil Sci.
2008;59:540-550. doi: 10.1111/j.1365-2389.2008.01016.x

Mallon CA, Poly F, LeRoux X, Marring I, vanElsas JD, Salles JF. Resource pulses can
alleviate the biodiversity-invasion relationship in soil microbial communities.
Ecology, 2015;96(4):915-926. doi: org/10.1890/14-1001.1

Dommelen AV, Vanderleyden J. Chapter 12: associative nitrogen fixation. In: The
Biology of the Nitrogen Cycle. Elsiver B.V, 2007; pp.179-192. doi:
10.1016/B978-0-444-52857-5.X5000-0

Ali GS, Norman D, El-Sayed AS. Soluble and volatile metabolites of plant
growth-promoting rhizobacteria (PGPRs). Adv. Bot. Res. 2015;75:241-284. doi:
10.1016/bs.abr.2015.07.004

Results: Fig 10. What is the meaning the color in the correlation test.

Answer: Thank you for your professional comments. Black arrow represents soil
bacterial communities at the phylum level, red arrow represents soil nitrogen and
organic carbon fractions, and green arrow represents root morphology indices. Circle
and square denote the non-sterilized and sterilized soils, respectively. MBC,
microbial biomass carbon; SOC, soil organic carbon; TN, total nitrogen; C/N, ratio
of total soil organic carbon to TN; POC, particulate organic carbon; WSOC,
water-soluble organic carbon; NH4+-N, ammonium nitrogen; NO3--N, nitrate nitrogen;
NO2--N, nitrite nitrogen; RV, root volume; RS, root surface; RL, root length; RB,
root biomass. Please see Fig 8.

Discussion: Line 442-444: This paragraph is not clear, please rewriting.

Answer: We rewrote this paragraph in order to make it clear. Please see Line 506-516.
The detailed revision was showed as follows:

“The net SOC balance depends on the difference between new C gain and native SOC loss
[73]. Glucose addition at low level leads to native SOC loss approximately equal to
SOC gain in non-sterilized soil (Fig 2). Thus, the balance between accumulation and
loss of SOC is probably associated with priming effect [21]. Glucose input to soil
could activate dormant microorganisms, causing SOM decomposition [74]. But, new SOC
formation derived from glucose C could offset native SOC loss in the sterilized soil
with low level of glucose addition. Moreover, the residual rate of glucose-C in the
sterilized soil was higher than that in non-sterilized soil under the same level of
glucose addition (Fig 1). These results demonstrated that soil sterilization may
promote glucose-C sequestration in a low C soil. The process of autoclaving could
destroy soil structure and lead to soil particulate finer [75, 76], which enhances
available surface area and provide many sorption sites for glucose-C retention in
soil [77].”

Discussion: Line 444: The authors should not affirm that the glucose increased the
“net C sequestration”.

Answer: Thanks for your professional comments. The term of “net C sequestration” is
not seriously used in this sentence. We changed the “net C sequestration” into “net
SOC balance” in Line 506. In this research, net SOC sequestration equaled to the
difference between native SOC and new SOC derived from glucose-C.

Discussion: Line 477-479: How the addition and sterilization increase the indices of
root morphology. I consider this result contradictory. 

Answer: Thanks for your suggestions. We really found that the glucose addition and
sterilization increased the indices of root morphology and enhanced root biomass.
The possible reasons are: The glucose addition and/or soil sterilization
significantly increased the indices of root morphology. This was mainly associated
with the increase in the relative abundance of Burkholderia (Proteobacteria),
Paenibacillus (Firmicutes) and Streptomyces (Actinobacteria) by glucose addition
and/or soil sterilization. The bacterial phyla of Proteobacteria, which consists
mostly of G- bacteria and diazotrophs, classified as plant growth promoting
rhizobacteria (PGPR) [97, 98]. This PGPR exerts a beneficial effect on root growth
on the production of phytohormone (e.g. IAA) or on nutrients uptake (e.g. N) by
plants [99]. Some the other genera, such as Streptomyces, also have PGPR traits to
hydrolyse chitin [100]. The genus Paenibacillus is easily isolated from the
agricultural soil and rhizosphere [101] and has the characterise against pathogens
[102]. Therefore, the above mentioned genera improved root morphology and promoted
plant growth due to root hormones levels producing IAA in the rhizosphere and the
nutrients availability. Those suggested the positive synergistic effect on root
morphology between glucose addition and soil sterilization. Please see Line
566-577.

Discussion: Line 490-491: The authors should cite the similar studies mentioned.

Answer: We checked the references and substituted the references with the similar
studies mentioned. Please see Line 551. The following references were showed as
follows:

[92] Berendsen R L, Pieterse C, Bakker P. The rhizosphere microbiome and plant
health. Trends Plant Sci. 2012;17(8):478-486. doi: 10.1016/j.tplants.2012.04.001

Conclusions: The data presented does not permit conclude on changes in bacterial
richness when applied glucose or/and sterilize the soil.

Answer: Thanks for your professional comments. We added the data about the diversity
and richness indices of soil bacterial community in Results section, Table 1 and Fig
3. “The values of Ace, Chao and Shannon indices were increased with the glucose
addition levels, whereas the Simpson index showed the opposite trend. Compared with
non-sterilized soil, sterilized soil with high level of glucose addition increased
the values of Ace, Chao and Shannon indices by 13.8%, 7.9% and 11.4%, respectively;
and sterilized soil with low level of glucose addition increased them by 17.3%,
13.8% and 2.6%, respectively.” and “The PcoA plot based on the clr-transformed data
was used to presented the changes of bacterial community structures in soil. As
shown in Fig 3A, the total of 78.9% of variations in the composition of bacterial
communities could be explained by the first two principal components. The PCoA1
clearly separate glucose addition samples from without addition samples. The CK and
SS treatments were differentiated along the PcoA2, which demonstrated that soil
sterilization was a main determinant on the second principal coordinate. For the
same addition level, no-sterilized and sterilization soils formed a subgroup. As
presented by the hierarchical cluster analysis, bacterial communities revealed two
clusters comprising samples from all treatment groups (Fig 3B). the samples of CK
and SS were closer to each other and the samples of Glu-1, SS+Glu-1, Glu-2 and
SS+Glu-2 were closer to each other.” 

So, we rewrote the conclusion that “Glucose addition combined with soil sterilization
not only increased the SOC content and new SOC formation derived from glucose C, but
also increased the alpha and beta diversities of soil bacterial communities.
Although soil microbial communities were similar between non-sterilization and
sterilization, soil sterilization mainly increased the relative abundances of
Proteobacteria, Firmicutes and Verrucomicrobia at the phyla level. Furthermore, the
glucose addition, especially combined with soil sterilization improved root
morphology, promoted the potential abilities of root N metabolism, and increased the
amino acid synthesis in root. Overall, these results suggested the supply of C
substrate with heathy soil conditions well shapes soil microbial communities and
root morphology, and potentially increases soil C sequestration in agroecosystems.
However, the complexity of C substrate drives the function and structure of soil
microbial communities, which could lead to dynamics of plant growth and soil
nutrient transformation. Further research should be focused on the coupled mechanism
among nutrients transformation, plant growth and soil C sequestration under supply
of complicated C substrates for low C soil.”. Please see Line 616-628.

Other revisions:

We updated references according to the added statement in the whole manuscript.

We hope these changes address all the concerns that you have with this manuscript. We
look forward to your reply.

Sincerely,

Bianbin Qi and Deguo Lv on behalf of Sijun Qin

to Reviewers.docx
---

## [Decision Letter · Decision Letter 1]

2 Dec 2021

PONE-D-21-22234R1Glucose addition promotes
C fixation and bacteria diversity in C-poor soils, improves root morphology, and
enhances key N metabolism in apple rootsPLOS ONE

Dear Dr. Qin,

Thank you for submitting your manuscript to PLOS ONE. After careful consideration, we
feel that it has merit but does not fully meet PLOS ONE’s publication criteria as it
currently stands. Therefore, we invite you to submit a revised version of the
manuscript that addresses the points raised during the review process.

Please include the following items when submitting your revised
manuscript:A rebuttal letter that responds to each point raised by the academic
editor and reviewer(s). You should upload this letter as a separate file
labeled 'Response to Reviewers'.A marked-up copy of your manuscript that highlights changes made to the
original version. You should upload this as a separate file labeled
'Revised Manuscript with Track Changes'.An unmarked version of your revised paper without tracked changes. You
should upload this as a separate file labeled 'Manuscript'.If you would like to make changes to your financial disclosure,
please include your updated statement in your cover letter. Guidelines for
resubmitting your figure files are available below the reviewer comments at the end
of this letter.

We look forward to receiving your revised manuscript.

Kind regards,

Ying Ma, Ph.D.

Academic Editor

PLOS ONE

Journal Requirements:

Reviewers' comments:

Reviewer's Responses to Questions

**Comments to the Author**

1. If the authors have adequately addressed your comments raised in a previous round
of review and you feel that this manuscript is now acceptable for publication, you
may indicate that here to bypass the “Comments to the Author” section, enter your
conflict of interest statement in the “Confidential to Editor” section, and submit
your "Accept" recommendation.

Reviewer #2: All comments have been addressed

2. Is the manuscript technically sound, and do the data
support the conclusions?

Reviewer #2: Yes

3. Has the statistical analysis been performed
appropriately and rigorously? 

Reviewer #2: Yes

4. Have the authors made all data underlying the
findings in their manuscript fully available?

Reviewer #2: Yes

5. Is the manuscript presented in an intelligible
fashion and written in standard English?

Reviewer #2: Yes

6. Review Comments to the Author

Reviewer #2: The authors improved substantially the manuscript; however, they should
consider the following observations:

-The authors should refer to the increase of a-diversity and changes in bacterial
community structure (B-diversity. You should avoid the use of the phrase "increase
of B-diversity". Please, check throughout the manuscript.

- The authors sequenced V3-V4 16S rRNA region. They should refer to the bacterial
community. Please, avoid the use of the microbial community throughout the
manuscript.

- The authors should change “heathy soil” for “healthy soil”. Please, check
throughout the manuscript.

7. PLOS authors have the option to publish the peer
review history of their article (what does this mean?). If published, this will
include your full peer review and any attached files.

If you choose “no”, your identity will remain anonymous but your review may still be
made public.

**Do you want your identity to be public for this peer review?** For
information about this choice, including consent withdrawal, please see our
Privacy Policy.

Reviewer #2: No

---

## [Author Response · Author response to Decision Letter 1]

5 Dec 2021

Dear Editor and Reviewers,

Ref.: PONE-D-21-22234R1

Glucose addition promotes C fixation and bacteria diversity in C-poor soils, improves
root morphology, and enhances key N metabolism in apple roots

Responses to Comments

Reviewer #2: 

The authors should refer to the increase of α-diversity and changes in bacterial
community structure (β-diversity. You should avoid the use of the phrase "increase
of β-diversity". Please, check throughout the manuscript).

Answer: We checked throughout the manuscript and changed “increased the alpha and
beta diversities of soil bacterial communities” to “increased the alpha diversity
and changed bacterial community structure in soils” in Line 618-619. Thanks for your
suggestion.

The authors sequenced V3-V4 16S rRNA region. They should refer to the bacterial
community. Please, avoid the use of the microbial community throughout the
manuscript.

Answer: We checked the whole manuscript and changed “microbial community” to
“bacterial community” in Line 76, 83, 88 and 565. Thanks for your professional
comments.

The authors should change “heathy soil” for “healthy soil”. Please, check throughout
the manuscript.

Answer: Thanks for your careful revision. We have changed “heathy soil” to “healthy
soil” in Line 624. Sorry for our miswriting.

Responses to Journal Requirements

Answer: We thoroughly checked the reference list of the manuscript to ensure that it
is complete and correct. All the cited references in paper have not been retracted.
The reference NO. 65 in the previous version “You YL, Wang JB, Shi WS, Pan DM.
Extraction of genomic DNA of Litchi chinensis and optimizationof the RAPD reaction
system. Biotechnology Bulletin. 2010;4113-115” is published in a Chinses journal. We
changed this reference to “Gasic K, Hernandez A, Korban SS. RNA extraction from
different apple tissues rich in polyphenols and polysaccharides for cDNA library
construction” published in Plant Mol Biol Rep. 2004;22(4):437-438 (a SCI journal) in
order to make the citation rational. Please see Line 263.

We hope these changes address all the concerns that you have with this manuscript. We
look forward to your reply.

Sincerely,

Bianbin Qi and Deguo Lv on behalf of Sijun Qin

to Reviewers.docx
---

## [Decision Letter · Decision Letter 2]

3 Jan 2022

Glucose addition promotes C fixation and bacteria diversity in C-poor soils, improves
root morphology, and enhances key N metabolism in apple roots

PONE-D-21-22234R2

Dear Dr. Qin,

We’re pleased to inform you that your manuscript has been judged scientifically
suitable for publication and will be formally accepted for publication once it meets
all outstanding technical requirements.

Kind regards,

Ying Ma, Ph.D.

Academic Editor

PLOS ONE

Additional Editor Comments (optional):

Reviewers' comments:

Reviewer's Responses to Questions

**Comments to the Author**

1. If the authors have adequately addressed your comments raised in a previous round
of review and you feel that this manuscript is now acceptable for publication, you
may indicate that here to bypass the “Comments to the Author” section, enter your
conflict of interest statement in the “Confidential to Editor” section, and submit
your "Accept" recommendation.

Reviewer #2: All comments have been addressed

2. Is the manuscript technically sound, and do the data
support the conclusions?

Reviewer #2: Yes

3. Has the statistical analysis been performed
appropriately and rigorously? 

Reviewer #2: Yes

4. Have the authors made all data underlying the
findings in their manuscript fully available?

Reviewer #2: (No Response)

5. Is the manuscript presented in an intelligible
fashion and written in standard English?

Reviewer #2: Yes

6. Review Comments to the Author

Reviewer #2: (No Response)

7. PLOS authors have the option to publish the peer
review history of their article (what does this mean?). If published, this will
include your full peer review and any attached files.

If you choose “no”, your identity will remain anonymous but your review may still be
made public.

**Do you want your identity to be public for this peer review?** For
information about this choice, including consent withdrawal, please see our
Privacy Policy.

Reviewer #2: No

---

## [Editor Report · Acceptance letter]

10 Jan 2022

PONE-D-21-22234R2 

Glucose addition promotes C fixation and bacteria diversity in C-poor soils, improves
root morphology, and enhances key N metabolism in apple roots 

Dear Dr. Qin:

I'm pleased to inform you that your manuscript has been deemed suitable for
publication in PLOS ONE. Congratulations! Your manuscript is now with our production
department. 

Kind regards, 

on behalf of

Dr. Ying Ma 

Academic Editor

PLOS ONE